



# Future water temperature of rivers in Switzerland under climate change investigated with physics-based models

Adrien Michel[1,2], Bettina Schaefli[3], Nander Wever[4], Harry Zekollari[5,6,7], Michael Lehning[1,2], and Hendrik Huwald[1,2]

[1]School of Architecture, Civil and Environmental Engineering, École Polytechnique Fédérale de Lausanne (EPFL), Switzerland
[2]WSL Institute for Snow and Avalanche Research (SLF), Davos, Switzerland
[3]Institute of Geography & Oeschger Centre for Climate Change Research, University of Bern, Bern, Switzerland
[4]Department of Atmospheric and Oceanic Sciences, University of Colorado Boulder, Boulder, USA
[5]Laboratory of Hydraulics, Hydrology and Glaciology (VAW), Eidgenössische Technische Hochschule (ETH), Zürich, Switzerland
[6]Swiss Federal Institute for Forest, Snow and Landscape Research (WSL), Birmensdorf, Switzerland
[7]Department of Geoscience and Remote Sensing, Delft University of Technology, Delft, the Netherlands

**Correspondence:** Adrien Michel (adrien.michel@epfl.ch)

**Abstract.**

Rivers are ecosystems highly sensitive to climate change and projected future increase in air temperature is expected to increase the stress for these ecosystems. Rivers are also an important socio-economical factor. In addition to changes in water availability, climate change will impact the temperature of rivers. This study presents a detailed analysis of river temperature and discharge evolution over the 21st century in Switzerland, a country covering a wide range of Alpine and lowland hydrolog-

ical regimes. In total, 12 catchments are studied. They are situated both in the lowland Swiss Plateau and the Alpine regions and cover overall 10% of the country's area. This represents the so far largest study of climate change impacts on river temperature in Switzerland. The impact of climate change is assessed using a chain of physics-based models forced with the most recent climate change scenarios for Switzerland including low, mid, and high emissions pathways. A clear warming of river water

is modelled during the 21st century, more pronounced for the high emission scenarios and toward the end of the century. For the period 2030-2040, median warming in river temperature of +1.1°C for Swiss Plateau catchments and of +0.8°C for Alpine catchments are expected compared to the reference period 1990-2000 (similar for all emission scenarios). At the end of the century (2080-2090), the median annual river temperature increase ranges between +0.9°C for low emission and +3.5°C for high emission scenarios for both Swiss Plateau and Alpine catchments. At the seasonal scale, the warming on the Swiss Plateau

and in the Alpine regions exhibits different patterns. For the Swiss Plateau, the spring and fall warming is comparable to the warming in winter, while the summer warming is stronger but still moderate. In Alpine catchments, only a very limited warming is expected in winter. A marked discharge increase in winter and spring is expected in these catchments due to enhanced snowmelt and a larger fraction of liquid precipitation. Accordingly, the period of maximum discharge in Alpine catchments, currently occurring during mid-summer, will shift to earlier in the year by a few weeks (low emission) or almost two months

(high emission) by the end of the century. In summer, the marked discharge reduction in Alpine catchments for high emission



scenarios leads to an increase in sensitivity of water temperature to low discharge, which is not observed in the Swiss Plateau catchments. In addition, an important soil warming is expected due to glacier and snow cover decrease. These effects combined lead to a summertime river warming of +6.0°C in Alpine catchments by the end of the century for high emission scenarios. Two metrics are used to show the adverse effects of river temperature increase both on natural and human systems. All results
of this study along with the necessary source code are provided with this manuscript.

## 1   Introduction

River systems are considered to be among the ecosystems most sensitive to climate change (CC) (Watts et al., 2015) and the projected future increase in air temperature (IPCC, 2013) are expected to increase the stress for these ecosystems. Water temperature is one of the most important variables for aquatic ecosystems, influencing both chemical and biological processes
(Benyahya et al., 2007; Temnerud and Weyhenmeyer, 2008). Certain fish species are highly sensitive to warm water, which can promote specific diseases (e.g. proliferative kidney disease, PKD) or prevent reproduction (Caissie, 2006; Carraro et al., 2016). In Alpine regions, along to water temperature, glacier retreat will also contribute to accelerate changes in ecosystems (Cauvy-Fraunié and Dangles, 2019; Fell et al., 2021). Higher temperatures might be favorable for some species, such as macro-invertebrates, enhancing biological invasion, which has already been observed (Paillex et al., 2017; Niedrist and Füreder, 2021).
In general, river temperature rise is expected to lead to a shift of many species' habitat to higher elevations. However, man-made or natural barriers (e.g. dams, power plants, waterfalls) might prevent migration to thermal refuges further upstream. In this context, it is necessary to underline that Alpine streams offer environmental heterogeneity and host a variety of species and genetic diversity; changes in water availability and in temperature will have an impact on the biodiversity and the ecosystem services at all levels.
River temperature is an important socio-economical factor. The literature clearly identified several sectors which are vulnerable: agriculture, tourism, electricity production, and drinking water supply and quality (e.g. Hock et al., 2005; Barnett et al., 2005; Schaefli et al., 2007; Bourqui et al., 2011; Viviroli et al., 2011; Beniston, 2012; Hannah and Garner, 2015). For example, during the exceptional heat wave and dry period in central and northern Europe from April through August 2018, local electricity production at the Swiss nuclear power plant Mühleberg, canton Bern, had to be temporarily reduced due to
unusually high water temperature of the Aare river.

River systems are considered to be among the ecosystems most sensitive to climate change (CC). Increase in surface water temperature is expected to affect ground water temperatures that are fed by river infiltration, with significant consequences on the biochemistry of these reservoirs (Epting et al., 2021). However, the dynamics of groundwater temperature are complex, and changes in precipitation patterns might lead to a cooling of ground water due to shifts in recharge periods toward colder seasons (Epting et al., 2021). For all the above reasons, quantitative information on the future evolution
of river temperature is essential and necessary. This study attempts to provide such information and predictions on the basis of different emission scenarios.

River temperature is expected to be affected by CC mainly through the influence of rising air temperature, changes in precipitation, and changes in snow and ice melt. At global scale, several studies have shown a clear trend in river temperature



at various locations over the last decades (Morrison et al., 2002; Webb and Nobilis, 2007; van Vliet et al., 2013; Null et al., 2013; Ficklin et al., 2014; Hannah and Garner, 2015; Watts et al., 2015; Santiago et al., 2017; Dugdale et al., 2018; Jackson et al., 2018), as well in lake surface temperature (Dokulil, 2014; O'Reilly et al., 2015; Woolway and Merchant, 2017; Woolway et al., 2020a, b). In Switzerland, a recent study shows a mean increase of river temperature of $0.33 \pm 0.03$ °C per decade between

1980 and 2018, which is mainly attributed to the increase in air temperature (Michel et al., 2020). This research also shows that the response to CC in Alpine catchments is different than in Swiss Plateau (lowland) catchments. The comparison of the future evolution of lowland versus Alpine catchments is one of the focal points of the present work.

Studies investigating the future evolution of water temperature in Switzerland and the Alps in general are sparse. For instance, using the CH2011 (CH2011, 2011) A1B CC scenario and a conceptual runoff model along with a simplified physics-based

model for water temperature, Råman Vinnå et al. (2018) find an increase of +0.08 °C per decade for the Rhône river and of +0.10 °C per decade for the Aare river throughout the 21st century. For simulating future river temperature evolution, a wide range of existing hydrological models is available (see the discussion in the review of Horton et al., 2021). Systematic reviews of such models exist in the literature (e.g. Benyahya et al., 2007; Gallice et al., 2015). These models are generally grouped into two main families: statistical or physics-based models. Statistical models rely on statistical relations to reproduce

the variables of interest based on available input data. They present different level of complexity from linear relationships (see e.g. Hrachowitz et al., 2010; Segura et al., 2015) to machine learning (see e.g. Feigl et al., 2021). They require an extensive calibration phase (or learning phase in the case of machine learning models), and calibrated models might not be valid outside of the observed temperature range, which is an important drawback in case of climate change studies (Benyahya et al., 2007).

On the other hand, physics-based models use a physical formulation of the mass or energy conservation to simulate discharge

or temperature. These models generally require more input data than statistical models and are significantly more demanding in terms of computation. There are only a few coupled physics-based hydro-thermal models resolving discharge and temperature at the same time; a review of these models can be found in the work of Gallice et al. (2016). Most of these coupled models use statistical degree-day methods to simulate snowmelt. However, a more physics-based representation of the snow processes in space and time, despite requiring usually more input data, allows for improved snow-runoff modelling during snowmelt

season, which is crucial in Alpine catchments (Martin and Etchevers, 2005; Magnusson et al., 2011; Lisi et al., 2015; Brauchli et al., 2017).

In this study we use the snowmelt and -runoff model Alpine3D (Lehning et al., 2006) coupled to the the semi-distributed hydrological model StreamFlow (Gallice et al., 2016). This model chain has already been successfully applied to Alpine hydrological modelling by Comola et al. (2015), Wever et al. (2017), Brauchli et al. (2017), and Griessinger et al. (2019).

Discharge is a key variable required for simulating the water temperature of streams, and it will be considered alongside stream temperature. Discharge is however not the main focal point here and some recent studies on discharge evolution in Switzerland with climate change over a larger set of catchments already exist (Brunner et al., 2019a, b; Muelchi et al., 2020, 2021). In the present study, the model chain is run over 12 catchments covering together 10% of the area of Switzerland; 7 are located in the Swiss Plateau, and 5 in the Swiss Alps. They cover a wide range of catchment sizes (from 3.4 to 973 km$^2$, see Figure 1).

The models are run for 5 periods of ten years between 1990 and 2090 and forced with an hourly downscaled version of the



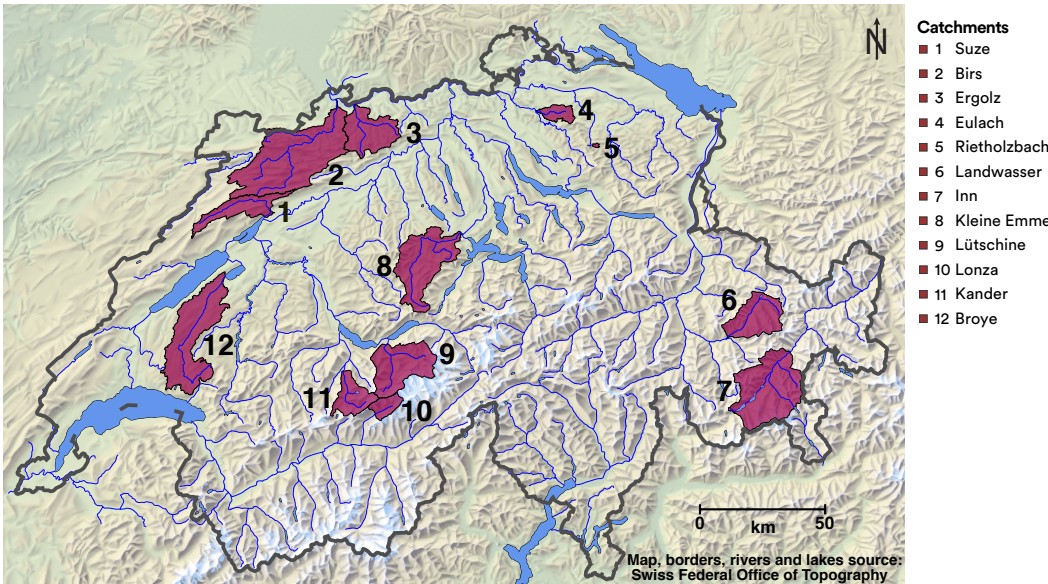

**Figure 1.** Map of Switzerland showing the location of the simulated catchments. Maps providing details of individual catchment are shown in Supplementary Figure S1. Data provided by the Swiss Federal Office of Topography (swisstopo).

CH2018 climate change scenarios (MeteoSuisse et al., 2018b; Michel et al., 2021b). The time series obtained from the models are analysed on monthly to annual time scales to describe and investigate the impact of CC. Some key indicators regarding ecology and industrial water usage are also computed, similar to the ones computed in the work of Michel et al. (2020) over a historical period.

## 2 Data

### 2.1 Catchments

For this study, 12 catchments are selected; they are shown in Figure 1 and their characteristics are listed in Table 1. The objective is to have catchments both on the Swiss Plateau and in the Swiss Alps. The selection of catchments is based on the following requirements: Sufficient availability of hydrological and meteorological measurements (Sections 2.2, 2.3) and of climate change scenarios (Section 2.4). Moreover, catchments with minimal anthropogenic disturbances and without important lakes along the watercourse are chosen since these effects are not taken into account in the models (Section 3.3). These criteria strongly reduce the number of candidate Alpine catchments, resulting in the following set: Inn, Kander, Landwasser, Lonza and Lütschine.

For the Swiss Plateau, more catchments satisfy the requirements. Considering a range of catchment sizes, the following catchments are retained: Birs, Broye, Ergolz, Eulach, Kleine Emme, Rietholzbach, and Suze. This selection is also based on





**Table 1.** Details of the selected catchments. Details about discharge stations (Q station) and water temperature stations (T station) are given in Table S1. Details about meteorological stations are given in Table S2. Mean annual water temperature and discharge are given at the gauging stations, which do not necessarily correspond to the points simulated with climate change scenarios (see text). They are computed over the period 2005-2015, except for the Ergolz (2014-2018).

| Catchment | Area (km²) | Mean elevation (m) | Min–Max elevation (m) | Glacier cover (%) | Mean annual discharge (mm h⁻¹) | Mean annual water temperature (°C) | Q station | T station | Meteo stations |
|---|---|---|---|---|---|---|---|---|---|
| **Swiss Plateau Catchments** | | | | | | | | | |
| Birs | 973.4 | 747 | 257–1436 | 0 | 1.57 | 11.0 | 2106 | 2106 | BAS CHA RUE |
| Broye | 627.3 | 667 | 429–1509 | 0 | 1.47 | 11.2 | 2034 | 2034 | CDF NEU PAY |
| Ergolz | 301.3 | 564 | 261–1151 | 0 | 1.11 | 12.0 | 2202 | ER1 | BAS BUS RUE |
| Eulach | 74.2 | 535 | 410–884 | 0 | 0.95 | 10.7 | ZH523 | ZH523 | KLO SMA TAE |
| Kleine Emme | 479.9 | 1053 | 436–2319 | 0 | 2.7 | 9.6 | 2634 | 2634 | LUZ NAP PIL |
| Reitholzbach | 3.4 | 794 | 672–927 | 0 | 2.5 | 8.5 | 2414 | 2414 | TAE STG |
| Suze | 214.9 | 985 | 432–1602 | 0 | 2.1 | 9.0 | A024 | A024 | CDF CHA NEU |
| **Alpine Catchments** | | | | | | | | | |
| Inn | 625.2 | 2463 | 903–4029 | 6.4 | 2.8 | 5.0 | 2462 | 2462 | COV SAM BER2 BER3 KES2 ZNZ2 |
| Kander | 180.2 | 2139 | 774–3662 | 13.3 | 4.2 | 6.8 | A017 | A017 | ABO INT JUN |
| Landwasser | 295.4 | 2134 | 958–3127 | 0.2 | 1.5 | 4.3 | 2355 | 2327 | DAV WFJ DAV2 DAV3 DAV4 KLO2 PAR2 SLF2 ZNZ2 |
| Lonza | 78.6 | 2619 | 1513–3864 | 26.4 | 5.1 | 4.1 | 2269 | 2269 | ABO INT JUN VIS GAN2 |
| Lütschine | 384.7 | 2032 | 575–4121 | 14.7 | 4.2 | 6.0 | 2109 | 2109 | ABO INT JUN LHO2 SCH2 |

the catchments of interest for groundwater studies, for which results from this study are integrated in the work of Epting et al. (2021).

For some catchments, the simulations are extended further downstream of the hydrological gauging station used for calibration, to allow for connection to lakes, and to cover areas required for the study performed in the work of Epting et al. (2021). A
5  detailed map showing the topography, catchment boundaries, stream network, and locations of hydrological and meteorological stations for each catchment is shown in Figure S1.

## 2.2 Hydrological data

Quality controlled water temperature and discharge measurments at hourly resolution are provided by the Federal Office for the Environment, FOEN (FOEN, 2019), the Office for Water and Waste of the Canton of Bern, AWA (AWA, 2019), the Office for
10  Waste, Water, Energy and Air of the Canton Zurich, AWEL (AWEL, 2019) and by Holinger AG. Details about the hydrological stations are given in Table S1 and Figure S1.



## 2.3 Meteorological data

Meteorological data used in this study are provided by the MeteoSwiss (MCH) automatic monitoring network, distributed through IDAWEB (2020), and by the Inter-Cantonal Measurement and Information System (IMIS, 2019). Each catchment is simulated using forcing data from 2 to 9 IMIS and MCH stations, depending on the number of available stations in or

nearby the catchment (for details see Tables 1, S2 and Figure S1). The variables used to force the model are: air temperature (TA), precipitation accumulation (PSUM), wind velocity (VW), relative humidity (RH) and incoming shortwave solar radiation (ISWR) at hourly resolution. Only variables that are available at measurement stations and in the downscaled CH2018 dataset (see Section 2.4) are used, to ensure that the historical and climate change runs use exactly the same set of forcing data.

Note that stations of the IMIS network do not provide ISWR. In addition, IMIS stations are not equipped with heated

precipitation gauges. For these stations, precipitation is deduced from snow depth variations during the winter season using the snow settling calculated by the SNOWPACK model (Lehning et al., 2002b) and from interpolation from nearby MCH stations with heated rain gauges in case of absence of snow.

Incoming longwave radiation (ILWR) is also required to force the models (see Section 4) and is measured at some MCH stations. However, this variable is not included in the CH2018 dataset used to force the model during climate change simulations.

As a consequence, both for historical and climate change periods, ILWR is calculated at the location of the meteorological stations applying an "all sky" approach described in the work of Omstedt (1990), which uses TA, RH and ISWR to estimate the cloud cover fraction and the longwave downward radiation.

## 2.4 Climate change scenarios

Recent climate change scenarios are available for Switzerland from the CH2018 dataset (MeteoSuisse et al., 2018a) at daily

resolution; for the purpose of this study a downscaled version of this dataset at hourly resolution has been produced, including an extension of the CH2018 scenarios to the IMIS station network (Michel et al., 2021a). The temporal downscaling is done using a delta change approach which is shown to correctly preserve the seasonal means of the CC scenarios. Since the requirements of this method regarding the length of the historical timeseries are stricter than the requirements of the procedure used to derive the CH2018 scenarios (i.e. longer historical timeseries are required), some stations are excluded from the original

dataset, reducing the numbers of available stations. In addition, the downscaling method used enforces an analyse of results ate monthly or seasonal scale. Shortest time periods are not correctly captured (see discussion in the work of  Michel et al., 2021b).

Most IMIS stations were installed in the years after 2000, such that only 10 years periods of downscaled CC scenarios can be constructed (see Michel et al., 2021b). For the IMIS stations, the temporally downscaled dataset for CC scenarios is

computed for each individual decade and for all stations between 1990 and 2100. For the MCH stations, which generally have much longer data availability, scenarios for 30 years periods between 1980 and 2100 were also constructed along with 10 years periods. Using the time series derived over 30 years would be beneficial since 30 years periods are generally considered to capture the climatic cycles better than 10 years periods (Michel et al., 2021b) and is often the standard length for CC studies





**Table 2.** Climate change model chains used in this study. For each model chain RCP2.6, RCP4.5 and RCP8.5 are used.

| GCM | RCM | Seed | Resolution |
|-----|-----|------|------------|
| ICHEC-EC-EARTH | DMI-HIRHAM5 | r3i1p1 | 11° |
| ICHEC-EC-EARTH | SMHI-RCA4 | r12i1p1 | 11° |
| MIROC-MIROC5 | SMHI-RCA4 | r1i1p1 | 44° |
| MOHC-HadGEM2-ES | KNMI-RACMO22E | r1i1p1 | 44° |
| MOHC-HadGEM2-ES | SMHI-RCA4 | r1i1p1 | 44° |
| MPI-M-MPI-ESM-LR | SMHI-RCA4 | r1i1p1 | 44° |
| NCC-NorESM1-M | SMHI-RCA4 | r1i1p1 | 44° |

(WMO, 2017). However, this would prevent the usage of IMIS stations. Magnusson et al. (2011) and Schlögl et al. (2016) have shown that increasing the number of stations used to force the model does improve the simulations over Alpine catchments. In addition, the computational limits of the models used here prevent simulations of 30 years periods. Accordingly, we use the 10 years time series in this work. In Section 4.3 we assess the impact of using 10 years compared to 30 years periods.

Out of the 68 CC scenarios provided in the work of Michel et al. (2021b), 21 are used in the present study: 7 for the RCP2.6 emission scenario (low to negative emission), 7 for the RCP4.5 emission scenarios (moderate emission), and 7 for the RCP8.5 scenarios (business-as-usual). Climate change scenarios originate from 7 chains of GCMs and RCMs detailed in Table 2. Only these 7 model chains contain all the variables and RCPs needed for the simulations performed here. The CC simulations are run over the hydrological years 1991-2000, 2006-2015, 2031-2040, 2056-2065, and 2081-2090, referred to as CC periods. For

simplicity, we use full decade names further in this paper (e.g. 1990-2000 for the hydrological years 1991-2000, meaning 1 Oct. 1990 to 30 Sept. 2000). The period 2005-2015 is used to validate the CC simulations versus measurements for catchments (when enough historical measurements are available, Section 4.2), this period thus is also referred to as CC validation period.

## 2.5    Elevation, glacier, catchment geometry, and land cover data

Multiple geographical data are needed to perform the simulations with Alpine3D. A digital elevation model is needed, as well

as a land use classification to initialize the pixels in the model in an appropriate state and define the soil and canopy properties. For glacierized catchments, the glaciated area and glacier thickness need to be provided.

   The digital elevation model (DEM) is derived from the DTM25 dataset at 25m resolution provided by Swisstopo, averaged to the resolutions used for the simulations (100 m and 500 m). Land cover data are derived from the 2006 version of the Copernicus CORINE Land Cover (European Environment Agency, 2013) dataset (CLC) at 100 m resolution (upscaled to

500 m resolution). CLC land cover classes are translated into the land cover classes available in Alpine3D (see Table S3). The catchment and hydrological network, together with sub-catchments attached to each river reach, are derived using the TauDEM software (Tarboton, 1997) with a wrapper to force it reproducing exactly the river network provided by the Swiss Federal Office





for the Environment (FOEN) (Swiss Federal Office of the Environment, 2013, 2020). Details about this method along with an evaluation are given in Section S2.

Detailed glacier thickness maps (i.e. ice thickness above bedrock surface) are used at the starting point for each past and future simulation. The evolution of the glacier geometry is simulated with the model GloGEMflow (Zekollari et al., 2019).

Details are presented in Section 3.1. Note that the glacier maps overwrite the CLC land cover classes, and that a pixel considered as glacier in CLC but not in the glacier model is turned into a bare rock pixel.

Glacier coverage and mean elevation indicated in Table 1 are obtained from the glacier height grids and DEM described above, which means that they might differ slightly from values given by the data provider at the gauging stations.

## 3   Models

The models used in this study, GloGEMflow, Alpine3D and StreamFlow, are presented in detail in the work of Zekollari et al. (2019), Lehning et al. (2006) and Gallice et al. (2016), respectively. Here we only provide an overview of the models and emphasize aspects relevant for the present application. All model setup files for Alpine3D and StreamFlow, along with soil properties, DEM, land use, glacier cover, and stream and watershed delineation files are provided (see Data and code availability).

### 3.1   GloGEMflow

GloGEMflow calculates the evolution of all individual glaciers along their flowlines by explicitly accounting for both surface mass balance and ice flow processes. The mass balance is calculated from a positive degree-day approach (Huss and Hock, 2015), while ice flow is described through the shallow-ice approximation (Hutter, 1983). GloGEMflow was extensively evaluated over the European Alps, by relying on observed mass balances, surface velocities and glacier changes and by comparing

the simulated glacier changes to those from high-resolution 3D modelling studies that focus on individual glaciers (e.g. Jouvet et al., 2009; Zekollari et al., 2014). The simulated glacier extents under the CH2018 CC scenarios considered in this study were transformed from the GloGEMflow 1D model grid to the 2D model grid (at 100 m and 500 m resolution) by ensuring that the area and volume of each elevation band was conserved. This conversion was performed by taking the 2D reference glacier geometry (Huss and Farinotti, 2012) as a starting point, and applying a uniform absolute change in ice thickness per

elevation band to match the GloGEMflow modelled area. Subsequently, the resulting 2D ice thickness was changed uniformly (same relative change) per elevation band to match the modelled GloGEMflow volume.

### 3.2   Alpine3D

Alpine3D is a spatially distributed version of the multi-layer snow and soil model SNOWPACK, which explicitly solves the mass and energy balance equations and simulates the snow micro-structure (Lehning et al., 2002b, a). As discussed in the

introduction, previous studies have shown the added value of a complex snow model in Alpine environments, while we argue that for Swiss Plateau regions, such complex models may not be required. However, Alpine3D provides the vertically resolved





soil temperature, which is required in StreamFlow and not provided in simpler models. In addition, using Alpine3D throughout allows to have consistent modelling between all catchments. Alpine3D is run at 500 m resolution for all catchments except the small Rietholzbach catchment, where a resolution of 100 m is used. The resolution is chosen to reduce the computational cost and it has been shown to have only a minor impact on simulated snow depth (Schlögl et al., 2016). The input data for

Alpine3D are interpolated to the grids using various algorithms provided by the MeteoIO library (Bavay and Egger, 2014). The air temperature is first de-trended for elevation (using a vertical lapse rate computed from the measurements), then interpolated using an inverse distance weighting, and finally re-trended. An analog procedure is applied for longwave radiation (using a fixed lapse rate of -31.25 W m$^{-2}$ km$^{-1}$ to mimic the effect of decreasing air temperature), for wind velocity (using the lapse rate computed from the measurements) and for precipitation, where values of the vertical lapse rate range between 10 % km$^{-1}$ and

50 % km$^{-1}$ (see Section 4). Finally, cloud cover is derived at each meteorological station from ISWR and interpolated to the grids using an inverse distance weighting algorithm. This cloud cover is then used to correct the theoretical diffuse and direct radiation at each pixel (Helbig, 2009). Topographical shading is taken into account and a simple model of reflected radiation from surrounding terrain is used.

Alpine3D also contains a two-layer canopy module simulating the micro-meteorology in the forest, the evapotranspiration,

and the interaction between trees and snow, including snow interception (Gouttevin et al., 2015). Grass, crops and other land covers are not directly simulated by the canopy module, and the evapotranspiration here is parameterized through the value of the roughness length used in the computation of the latent heat flux. Water infiltration in snow and soil is handled through a simple bucket model. A more complex option of explicitly solving Richard's equation for water transport in the snowpack or in the soil is implemented in SNOWPACK and described in the work of Wever et al. (2014, 2015). These studies discussed that

the bucket scheme performs well on seasonal time scales, while Richard's equation, which provides more accurate results on sub-daily time scales or very early in the melt season, also comes at higher computation costs. This higher computational cost prevents the usage of this implementation in the large study performed here.

The model writes gridded output for all interpolated forcing variables along with the soil temperature at various depths and the runoff at the bottom of the soil column. Maps of snow depth and SWE are only partially used in the discussion here, but

might be of interest for further studies on the spatial impact of CC on snow depth in the future, and are therefore included in the model output.

### 3.3   StreamFlow

StreamFlow is a semi-distributed model simulating discharge and temperature at the same time in each river segment. The runoff output at the bottom of the soil column produced in Alpine3D is collected and summed up at the scale of each each

sub-catchment in StreamFlow, and the residence time in the soil is determined with an approach using two linear reservoirs in series (Comola et al., 2015), with reservoir coefficients to be calibrated. A parameter representing a fraction of water loss (representing deep soil infiltration or the difference between surface and subsurface catchment) can be calibrated in addition. A single parameter set is calibrated for the entire catchment, with the reservoir parameters being scaled to the sub-catchment size (Gallice et al., 2016).



Based on this sub-catchment runoff, the model then uses either a lumped approach (where each reach is resolved as one element, receiving input from its sub-catchment) or a discretized approach (where reaches are separated into sub-elements based on the resolution used) to compute reach-scale water temperature and runoff routing to the outlet.

For water routing at the reach scale or at the sub-element scale, either an instant routing is considered or a routing scheme
based on the Muskingum-Cunge approach, which solves a diffusive-wave approximation of the shallow water equation (Cunge, 1969; Ponce and Changanti, 1994). All details concerning this model are presented in the work of Gallice et al. (2016).

The water temperature in the soil reservoirs at the sub-catchment scale (which determines the water temperature when leaving the reservoirs and entering the river reaches) can be computed in StreamFlow either by (a) using the energy balance approach of Comola et al. (2015) (where one parameter needs to be calibrated), (b) using the approach in the Hydrological
Simulation Program–Fortran, HSPF (Bicknell et al., 1997), or (c) simply taking the soil temperature at a given depth. The HSPF approach essentially approximates the time evolution of the water temperature in the reservoirs by smoothing and adding an offset to the time series of air temperature (the smoothing factor and offset are calibrated parameters). For all three approaches, forcing values averaged over each sub-catchment are used. Different routing and soil water temperature schemes are tested for choosing the most suitable for this application (see Sections 4.1.1 and S5).

Once the water is routed to the river, the evolution of the water temperature is obtained by computing the energy balance for each reach (either lumped or gridded) considering solar and longwave radiations, sensible and latent heat fluxes, heat exchange with the streambed, friction with the ground, and heat advection from upstream reaches and from water infiltration from the stream-hillslope interface. The heat transfer coefficient between the ground and the river needs to be calibrated. In addition to the calibration of this heat transfer coefficient, the only other measurement-based model adjustment is the soil depth used
to determine the subsurface runoff temperature before entering a stream reach (except in HSPF scheme) and to compute the water/stream bed heat exchange. Different depths are tested (Sections 4.1.1 and S5). Both Alpine3D and StreamFlow are run at hourly resolution.

There is yet no possibility to include anthropogenic disturbances in StreamFlow such as water retention, pumping or energy input. Since dams are very abundant in Switzerland (Belletti et al., 2020; Mulligan et al., 2020), it reduces the choice of Alpine
catchments to be simulated.

## 3.4 Model code optimization

For Alpine3D as well as for StreamFlow, some optimization work was needed in order to use the model chain for such a computationally intense study. Indeed, Alpine3D is heavy in terms of computation. Even when run on multiple nodes with several cores each (36 on the CSCS supercomputer used here), a single run for 10 years at hourly resolution on an Alpine
catchment such as the Lütschine (385 km$^2$ and important snow cover) takes up to 18h in the current version (using 500 m resolution). Note that running 5 time periods for 21 CC scenarios means 105 runs per catchment. For StreamFlow, the heavy part in terms of computation is the calibration phase.

Substantial work has been done to optimize Alpine3D, especially in improving its numerical performance when run in parallel. To the best of our knowledge, this is the largest study, in terms of area covered, resolution and number of simulations,





ever made with Alpine3D. In StreamFlow, the calibration phase has been parallelized using OpenMP achieving a near-linear increase in performance with the number of CPUs used. The input module of StreamFlow has also been completely rewritten, significantly increasing the numerical performance. This allows to use longer calibration periods than in the work of Gallice et al. (2016) and to increase by one or two orders of magnitude the number of calibration runs (depending on the model setup).

Finally, many features have been added, such as the fraction of water loss in the soil, the KGE metric, or the possibility to perform the calibration using daily mean values instead of the internal hourly computation time step.

## 4   Models calibration and validation

### 4.1   Calibration and validation over historical period

This section presents the calibration and validation of the models. In the following, meteorological seasons abbreviated as

follows: Winter = DJF (December, January, and February), Spring = MAM (March, April, and May), Summer = JJA (June, July and August), and Fall = SON (September, October, and November). Sensitivity analysis on calibration period length is shown in Section S4, inter-comparison of different models setup for StreamFlow is shown in Section S5, and detailed figures for each catchment for the calibration and validation phase are shown in Section S6, Figures S4 to S27. Figures S28 to S31 show the snow depth of Alpine catchments simulated by Alpine3D at the location of stations measuring snow depth. Additional

plots for Alpine catchments calibration are shown in Section S7, Figures S32 to S37. To alleviate the text, we do not constantly refer to these figures in the rest of this section.

### 4.1.1   Calibration and validation setup

For the calibration/validation phases, Alpine3D is run for the hydrological years 2012-2018. Before formal parameter calibration in StreamFlow, multiple model runs of Alpine3D are performed with different values of the precipitation vertical lapse

rate in order to adjust the yearly total mass balance in Alpine catchments. In addition, modeled snow heights are compared to measurements to assess how well Alpine3D reproduces observed snow season dynamics in terms of season duration. Alpine3D has therefore undergone some parameter adjustment but cannot be calibrated as a physics-based model.

After this initial performance check of Alpine3D, StreamFlow is calibrated over the years 2012-2014 and validated over the years 2015-2018. An exception here is the Eulach catchment, where due to the lack of water temperature measurements before

2014, Alpine3D is run over 2015-2018, while the calibration and validation periods are 2015-2016 and 2017-2018. Section S4 shows sensitivity tests using the Broye and Lonza catchments and a longer simulation time period (2002-2018). Different calibration periods are used within these 17 years to test whether there is a strong influence on hydrological model output (Myers et al., 2021), which was not the case.

Depending on the setup, between 5 and 7 parameters need to be calibrated in StreamFlow (4 for discharge, the remaining

for water temperature, see Table 3). The calibration is completed with a Monte Carlo approach, first for the 4 parameters of the discharge module (50'000 runs), and then for the 1 to 3 parameters of the water temperature module (10'000 runs). The





**Table 3.** Calibration parameters and range of values used in StreamFlow, see the work of Gallice et al. (2016) for details.

| Parameter | Range | Units |
|---|---|---|
| **Discharge parameters** | | |
| Maximum infiltration rate | [0,100] | mm day$^{-1}$ |
| Upper reservoir $\tau$ | [1,50] | day |
| Lower reservoir $\tau$ | [100,1000] | day |
| Fraction of lost water | [0,40] | % |
| **Water temperature parameters** | | |
| Streambed heat transfer coefficient | [0,100] | W m$^{-2}$ K$^{-1}$ |
| Offset (HSPF module) | [-3,1] | s |
| Smoothing factor (HSPF module) | [1e-7,5e-6] | K s$^{-1}$ |
| Diffusion time (energy balance module) | [1e-3,100] | day |

calibration for water temperature is run only for the best parameter set obtained from discharge calibration, and is run for 6 different soil temperature depths from Alpine3D (from 0.5 to 3 m).

The sequential calibration is motivated by the fact that the model is significantly faster when only discharge is computed. The random sets are drawn from uniform distributions, with bounds indicated in Table 3 (taken and slightly adapted from the work of Gallice et al., 2016). All other parameters of the model, such as width-to-height relationship of the reach cross-section, are taken from the work of Gallice et al. (2016). As performance metrics, we use the Kling-Gupta efficiency (KGE) coefficient (Gupta et al., 2009) for discharge and the root mean square error (RMSE) for water temperature.

To assess the performance of the different StreamFlow modules, the calibration is performed using either the lumped or discretized, and the direct or Muskingum-Cunge approaches for reach-scale water routing (4 combinations), and the three sub-catchment temperature schemes. These are tested both at 100 m and 500 m resolution (24 combinations in total, Section S5). We find that using the 100 m resolution slightly improves the results, StreamFlow is thus run at 100 m resolution for the CC analysis. Note that even when using the lumped approach for routing, the StreamFlow simulation resolution impacts the delineation of sub-catchments and river reaches, and thus the simulated water and heat input to the river reaches. The more complex and computationally more demanding water routing schemes do not improve the performance, the lumped and direct approaches are thus used. Finally, the HSPF approach for sub-catchment temperature yielded the best results across all catchments and is therefore selected. Note that employing the HSPF scheme results in a lower impact of the soil temperature (only through conduction between water and streambed).

### 4.1.2 StreamFlow calibration and validation results

Table 4 shows the KGE values for discharge and the RMSE values for water temperature from calibration and validation of StreamFlow with the chosen settings. These values indicate a good performance in comparison to other results in the literature





**Table 4.** Performance of StreamFlow during the calibration and validation periods evaluated with Kling-Gupta efficiency (KGE) for discharge and RMSE for water temperature.

| Catchment | Calibration period | | Validation period | |
|---|---|---|---|---|
| | KGE (-) | RMSE (°C) | KGE (-) | RMSE (°C) |
| **Swiss Plateau Catchments** | | | | |
| Birs | 0.84 | 1.06 | 0.86 | 1.20 |
| Broye | 0.75 | 0.91 | 0.78 | 0.91 |
| Ergolz | 0.85 | 1.17 | 0.84 | 1.39 |
| Eulach | 0.74 | 1.18 | 0.67 | 1.08 |
| Kleine Emme | 0.79 | 1.08 | 0.70 | 1.07 |
| Rietholzbach | 0.74 | 1.63 | 0.75 | 1.81 |
| Suze | 0.84 | 1.68 | 0.87 | 1.50 |
| **Alpine Catchments** | | | | |
| Inn | 0.94 | 1.02 | 0.87 | 1.25 |
| Kander | 0.89 | 0.69 | 0.78 | 1.18 |
| Landwasser | 0.83 | 0.92 | 0.72 | 1.15 |
| Lonza | 0.92 | 0.89 | 0.91 | 1.01 |
| Lütschine | 0.89 | 1.28 | 0.84 | 1.37 |

(e.g. Köplin et al., 2010; Råman Vinnå et al., 2018). In particular, the RMSE values for water temperature are lower than in the work of Gallice et al. (2016) for an Alpine catchment (the Dischmabach catchment, part of the Landwasser catchment) using the first version of StreamFlow. All calibrated parameter values for each catchment are summarized in Table S8.

Time series of daily simulated and measured water temperature of four catchments are shown in Figure 2. The time series
show that catchments with similar performance metrics (see Table 4) can still show a different quality of fit to observed data over the year. This is most clearly visible by an overestimation of water temperature in Alpine catchments in summer, which underlines the limitation of using lumped model performance metrics such as KGE and RMSE over the entire year, and the need to perform a more detailed analysis, as we will discuss below.

**Swiss Plateau catchments**

Validation results for the plateau catchments show that the KGE ranges between 0.67 and 0.87 and the RMSE between 0.91 and 1.81 °C (see Table 4 and the detailed plots in Section S6). For these catchments, the validation time series for both discharge and water temperature lie in the range of the historical variability of the measurements. The dynamics of high water temperature and discharge events along with the annual cycles are well captured. There are no strong seasonal patterns for
the errors in water temperature in these catchments (except for a slight underestimation in spring), and there is no correlation between errors in discharge and water temperature. However, there is an overestimation of discharge in winter, but without an impact on simulated water temperature.





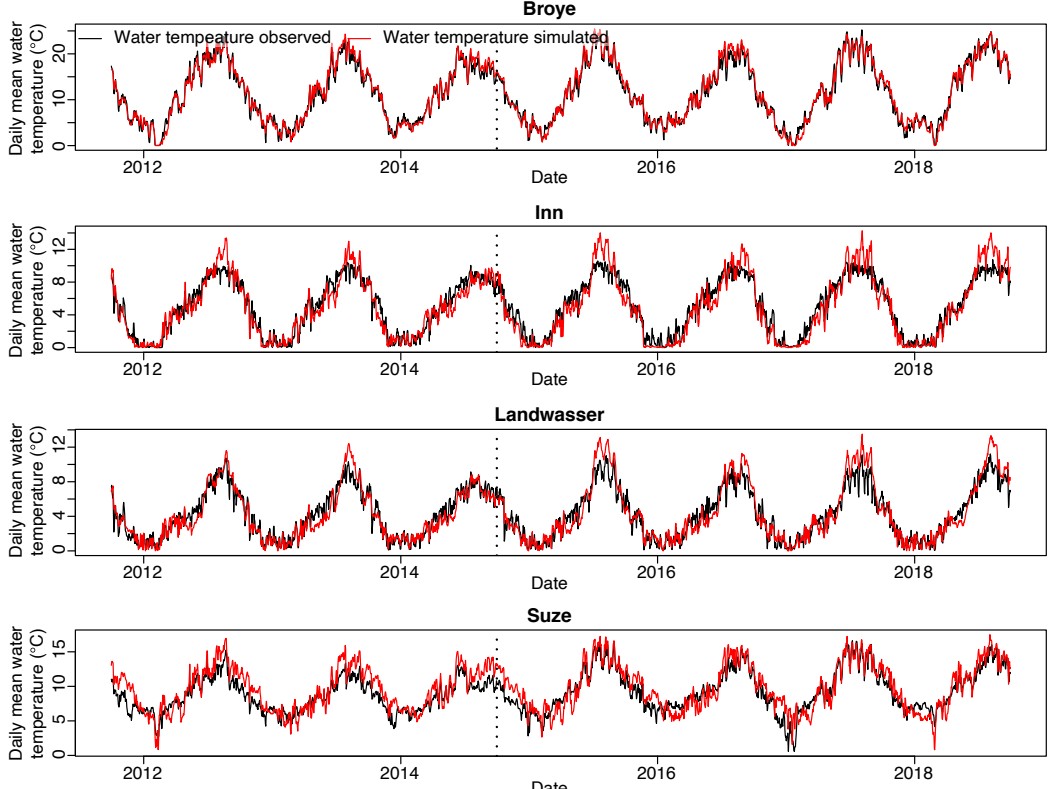

**Figure 2.** Water temperature observed (black) and simulated (red) over the calibration periods (left of the dashed line) and over the validation period (right of the dashed line) for four catchments: the Broye (Swiss Plateau), the Inn (Alpine), the Landwasser (Alpine), and the Suze (Swiss Plateau). These four catchments were chosen to represent the variation in the catchment type. Note that the extent of the y-axes (daily mean water temperature range) is different for every panel. Other catchments are shown in Section S6.

The error on temperature is slightly larger for the Suze catchment compared to the other Swiss Plateau catchments (see Table 4 and Figure 2). We attribute this to the fact that this region is karstic, with enhanced ground infiltration and resurgence to the surface, which impacts the water temperature. In addition, the gauging station is situated downstream of a cement factory, making anthropogenic disturbances in the measurements likely (see Michel et al., 2020). For the Eulach catchment, a large fraction

5  of water (about 33%) is directly lost to deeper groundwater via the model water loss parameter. However, this is coherent with the ratio of precipitation and discharge observed in this catchment as described in the work of Huggenberger and Epting (2011). Important soil water loss is also modelled for the Birs. Again this is not surprising since both the Birs and Eulach catchments have been selected specifically because of their interaction with ground water in order to be used in the study of Epting et al. (2021). Finally, the longer run performed for the Broye catchment (2002-2018, Figure S3) shows that the extremely warm

10  years 2003, 2015, and 2017 and the relatively cooler years 2007 and 2014 are well captured in the modelled water temperatures.





**Alpine catchments**

Alpine3D and StreamFlow perform very well in terms of snow cover and discharge for two of the Alpine catchments: The annual discharge cycle is well reproduced for the Kander (Figures S14, S15, and S29) and the Lütschine (Figures S22, S23, and S31). The results are less coherent for the Inn (Figures S12, S13, and S28) and the Lonza (Figures S20 and S21); this

is clearly visible from the discharge plots, but is not necessarily reflected in KGE values. For the Landwasser (Figures S18, S19, and S30), the melt season is clearly anticipated by Alpine3D, with a clear impact on water temperature, as shown by the negative correlation between discharge and water temperature errors in spring and summer. These points highlight the difficulty of accurately reproducing snow and glacier melt-induced runoff dynamics in Alpine environments, even when using a very sophisticated snow model. A possible explanation is the scarcity of meteorological measurements in Alpine regions,

which is expected to decrease the performance of the models (Magnusson et al., 2011).

For water temperature, lower model performances are obtained in summer for Alpine catchments compared to Plateau catchments. Sudden high water temperature peaks of up to $+4°C$ above the measurements are occasionally simulated in summer, leading to an error of up to $+2°C$ in the summer seasonal mean (Figure 2 and Section S6). This does not depend on the calibration period used. Since in Alpine catchments a larger water temperature increase is expected during future summers compared

to the Swiss Plateau catchments (see discussion below and in the work of Michel et al., 2020), this error needs to be understood.

This error appears in all Alpine catchments simulated here, regardless of a potential summer discharge underestimation. Indeed, the negative correlation between water temperature and discharge errors found for the Inn and the Landwasser are not obtained neither in the Lonza and nor for the Lütschine, meaning that the underestimation of summer discharge cannot explain the overestimated simulated water temperature.

We now discuss the Inn river in more detail as an example. Figure 3 presents the different energy fluxes at the outlet reach of the Inn river and highlights four events of critical water temperature overestimation. While all of these events correspond also to high soil temperature, there is no noticeable increase of the heat flux from the soil (panel h in Figure 3). Also, only the first event is directly related to an influx of energy from water originating from the surrounding catchment (panel i in Figure 3). By contrast, all four events are clearly correlated with a large amount of energy arriving from upstream reaches, suggesting the

issue is not caused by the local fluxes at the outlet, but rather by processes occurring upstream. Analysis of fluxes at two reaches situated upstream close to sources (Figures S31 and S32) shows that each of these events is related to high air temperature, but has a different origin. The first event is caused by warm water infiltration from the ground. The second event is caused by a combination of warm water infiltration, large downward latent heat fluxes (condensation) as well as sensible heat fluxes (convective heating of the water surface), and a large conductive heat flux from the streambed. The third event is caused by a

sudden increase of solar radiation (after a cloudy period) and a sudden increase in conductive energy flux from the streambed. Finally, the last event is related to very low discharge condition and thus higher sensitivity to the acting fluxes.

In summary, we did not find a single explanation for these warm events produced by the model. The analysis suggests that the upstream reaches are too sensitive to variations in the forcing, causing too high temperature in the upper part of the catchment, which then gets advected downstream. An analog analysis for the Inn and the Kander catchments yielded similar conclusions

(see Figures S34 to S37). The consequence of these errors are discussed in Section 4.4.

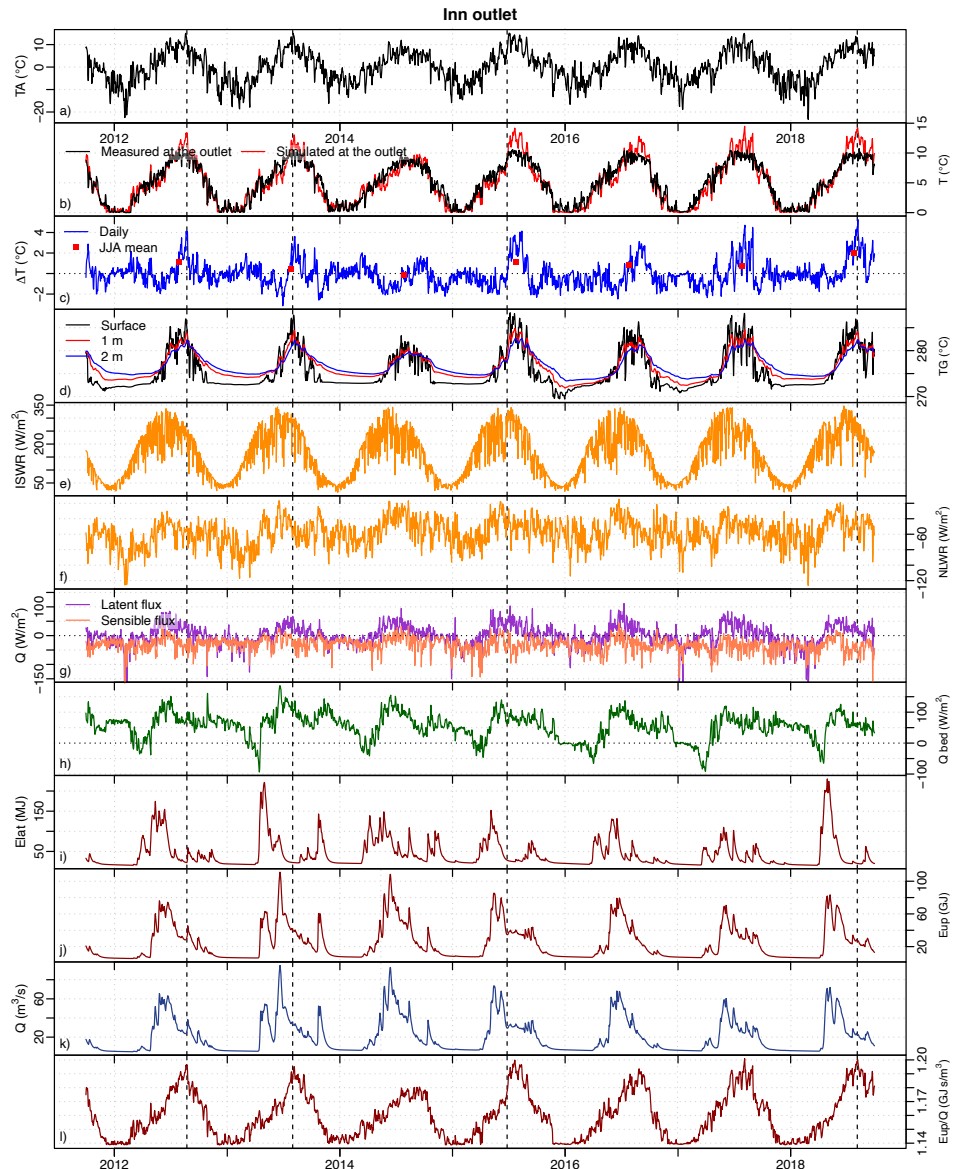

**Figure 3.** Energy fluxes and other relevant variables at the outlet reach of the Inn catchment. The vertical dashed lines mark some warm events discussed in the text. a) Air temperature, b) measured water temperature (black) and simulated water temperature (red) at the outlet, c) difference between simulated and measured water temperature (blue line), and mean over the difference over the summer season (red squares), d) temperature of the soil surface and at 1 m and 2 m depth, e) incoming shortwave radiation, f) net longwave radiation, g) latent heat flux (purple) and sensible heat flux (orange), positive means energy gain for the river, h) conductive heat flux between the bed and the river, positive means energy gain for the river, i) energy advected by water from the soil reservoirs, j) energy advected from upstream reaches, k) discharge, l) energy advected from upstream reaches normalized by discharge.





## 4.2 Validation over climate change periods

The models are run with all 21 CC scenarios over the period 2005-2015, and the discharge and water temperature obtained are compared to measurements over these periods. The model results are not expected to match the measurements at short timescale since CC scenarios are only supposed to represent the climatology of the considered period. The purpose of this verification is

to check whether the entire modeling chain (from CC scenarios to hydrological model) gives results which are, for the annual cycle, coherent with the measurements. The model outputs are thus compared to measurements in terms of annual cycle. The analysis is performed over all catchments (except for the Ergolz where not enough historical data are available) and results are shown in Section S8 (Figures S38 to S48).

For the Plateau catchments, the pattern of difference in temperature is similar to the error observed for the model validation

phase (compare e.g. Figures S6 and S39). For discharge, most of the difference between observed and modelled discharge is due to the difference in precipitation from CC simulations compared to measurements.

For the Alpine catchments, the overestimation of summer temperature is still present. In addition, the forcing CC models lead to over- or underestimation of the total discharge in Alpine catchments, depending on the catchment, but only to a limited extent. This is expected since the used CC scenarios show lower performances in Alpine areas than in Swiss Plateau areas for

precipitation (Warscher et al., 2019). Overall, it is confirmed that the output of Alpine3D and StreamFlow, when forced with CC scenarios, is coherent with historical time series.

## 4.3 Using 10 years versus 30 years time periods

To assess the impact of working with time series downscaled using only one decade of data instead of 30 years, we ran the models with climate change scenarios with both versions of the time series for the Broye and the Kleine Emme catchments.

The models are run both for the periods 1980-2010 and 1990-2000, and for the periods 2070-2100 and 2080-2090 with the selected 21 climate change scenarios, using either time series downscaled over 30 or 10 years period. Delta, i.e. difference between two time periods for a given variable, is computed between the future and past periods for water temperature and discharge using (a) the 10 years time series, (b) the 30 years time series, and (c) only 10 years in the middle of the 30 years time series, with forcing data downscaled over the 30 years period (Figures 4 and S49).

There are some obvious differences between the 3 runs introduced by the natural variability in the time series. There is also more spread in the delta values between scenarios when time series downscaled over only ten years are used, which is expected since time series downscaled over shorter periods are more prone to reflect model internal variability. However, the deltas obtained are very similar (Figures 4 and S49). Indeed, the median values of the deltas obtained with 10 years time series consistently lie within the range of deltas obtained using longer time series. This suggests that no significant and systematic

error is introduced when using the shorter time series. The main impact is an increase of the uncertainty in the results, due to the increased range of internal variability of the CC scenarios.



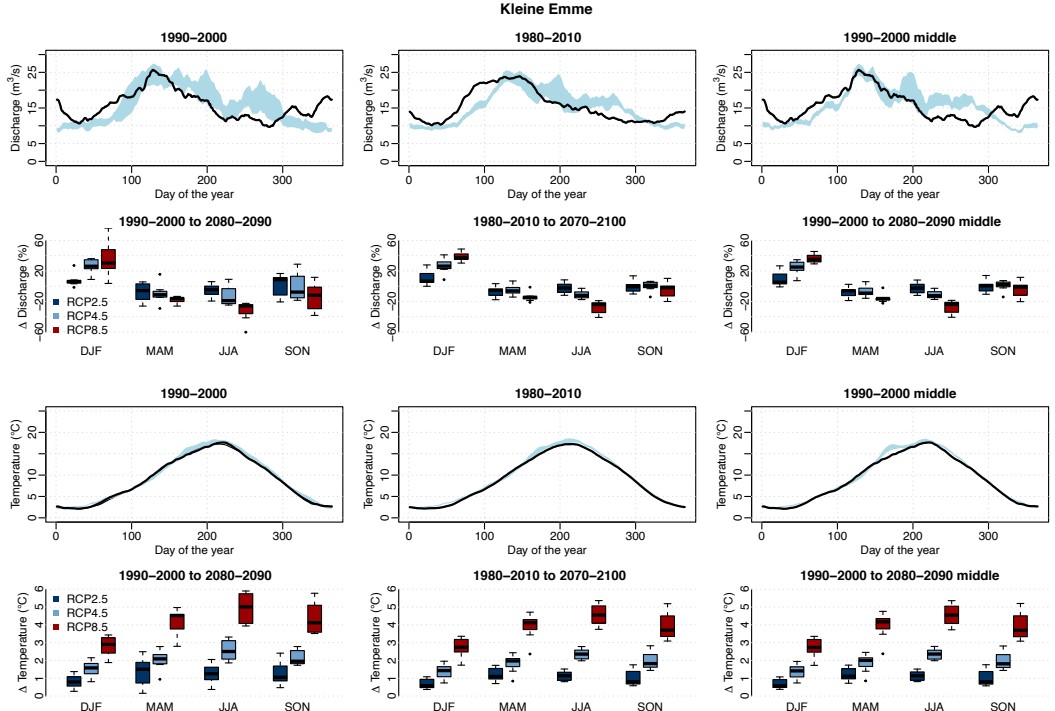

**Figure 4.** Impact of the version of the downscaled time series applied to the Kleine Emme catchment. Left: time series downscaled over 10 years periods and running the model over 10 years. Middle: time series downscaled over 30 years periods and running the model over 30 years. Right: time series downscaled over 30 years and running the model for the 10 years in the middle of the period. Row 1: Mean discharge over historical periods (1990-2000 or 1980-2010) for all 21 CC scenarios compared to measurements over the same period. Row 2: Change in mean discharge between end of the century periods (2080-2090 or 2070-2100) and historical periods grouped by RCP. Rows 3-4: Same as first two rows but for water temperature. For boxplots, boxes represent the first and third quartiles of the data, whiskers extend to points up to 1.5 time the box range (i.e. up to 1.5 time the first to third quartiles distance) and extra outliers are represented as dots.

## 4.4 Model performance in Alpine catchments

Michel et al. (2020) used historical measurements to show that large amounts of snow and glacier melt contribute to mitigate increased water temperature during warm summers; in their analysis, Alpine catchments in Switzerland were not affected by the extremely warm summer of 2003. In the current study, the model clearly predicts an important temperature increase for the
5   year 2003 in the high Alpine Lonza catchment where the model has been run for a longer time period (Figure S3).

This contradictory result is most likely due to a shortcoming in the current Alpine3D-StreamFlow model chain, which is the absence of an explicit parameterization of surface runoff processes. All water leaving the snowpack is assumed to infiltrate the soil at the pixel scale in Alpine3D, and the runoff at the bottom of the soil column is collected in StreamFlow at the sub-



catchment scale, after a transfer through the two linear reservoirs emulating fast and delayed lateral subsurface runoff. The water leaving these two reservoirs is assigned the temperature determined by the used sub-catchment temperature scheme.

While this is a fair approximation for the Swiss Plateau catchments, except for extreme precipitation events potentially generating Hortonian surface runoff or saturation-excess runoff (which are beyond the scope of this work), this is more problematic

for Alpine catchments. In particular, the occurrence of direct surface runoff input to the stream network, potentially occurring during snow and glacier melt on frozen or saturated soils, is not captured in the model; this can be critical because such surface runoff might advect very cold water to streams and bring the water close to 0°C.

In spring, the air and soil being still relatively cold, this has not too much impact on the simulated water temperatures. In summer, however, this cold water not being accounted for can explain the exaggerated sensitivity of small upstream reaches.

Another likely cause of the sensitivity overestimation in the model is the parameterization of the temperature of water leaving the soil and entering the river, obtained from quite simplistic formulations, along with the lack of dynamic interaction with the water table in the hyporheic zone as well as at the hillslope scale. In complex Alpine terrain, cold groundwater reserves are present, and they can contribute to mitigate the warm summer air temperature. However, on average, at catchment-scale, groundwater is known to be in equilibrium with the mean annual air temperature (Florides and Kalogirou, 2005; Epting and

Huggenberger, 2013), and this cooling effect can thus be expected to decrease in the future.

The cooling effect from snow and glacier melt is also expected to decrease in the future since in the European Alps the glacier melt peak water is occurring at present (Huss and Hock, 2018; Compagno et al., 2021). Any such transient effect related to glacier and seasonal snow cover retreat can only be partially captured in the model. In fact, in StreamFlow, this is only captured through the impact of reduced snow cover and glacier retreat on the soil temperature via removal of the insulation layer on

top of the soil and via a decrease of the albedo. Any additional effect via enhanced or reduced cool water input to the stream network is not explicitly parameterized.

In this study, the CC impact is assessed by looking at the delta between past and future time series; the above cooling effect from direct cold water input to the stream is clearly present in the past, but will most likely decrease or disappear by the end of the century for all catchments studied here. Accordingly, the lack of parameterization of cold water input can be expected to

actually result in an underestimation of the simulated future water temperature warming in Alpine catchments if the warming is obtained by a comparison with a past period.

To complete this analysis, it is however important to assess how the wrongly simulated hot spells in the past (resulting from a too high sensitivity of high elevation sub-catchments to heat input) translate into hot spells simulated under CC. This is particularly important because the water temperature is determined by many non-linear effects. We thus assess here how the

summer mean water temperature is evolving in the future to see if this error may be increasing over time.

For the Inn catchment, the difference between simulated summer water temperature in the cool summer 2014 and the warm summer 2015 is 4.8°C. Since the downscaled time series use the historical inter-annual variability over the period 2005-2015 as baseline data (see Michel et al., 2021b), the difference between the years 2014 and 2015 is also present in the CC time series (in the last two years of the used 10 year period). The model can thus be run with CC time series at the end of the century and

the difference between the two years can be compared to the historical difference between 2014 and 2015 (in terms of summer





average). Using the RCP8.5 scenarios for the decade 2080-2090, the mean difference of the 7 scenarios is of 6.7°C between these two years. A value of 5.8°C is obtained for the RCP2.6 scenarios.

The difference between warmer and cooler years is thus growing in the future, however in reasonable limits: as shown in Section 5.2, this will be accompanied by a marked decrease in discharge, especially with the RCP8.5 emission scenario,
further increasing the sensitivity to warm air temperature and contributing to the increase in the difference between warm and cold summers. Accordingly, the fact that the difference between cold and warm years remains reasonably low under RCP8.5 scenarios (despite of a mean air temperature increase of up to +8°C in summer for some scenarios compared to the reference period), supports our hypothesis that the model chain yields reliable estimates of future summer water temperature changes even for high Alpine catchments.

## 5 Climate change simulations – Results and discussion

This section presents the results of the CC simulations over the periods 2030-2040, 2055-2065, and 2080-2090. The results are shown in terms of changes (delta) compared to the reference period 1990-2000. Absolute change is used for water, soil and air temperature, and relative change for other variables. Detailed plots for each catchments are included in Supplementary Materials Section S9, Figures S50 to S61. The results for annual mean temperature for all catchments are summarized in Table
5. Detailed annual and seasonal values for all catchments are presented in Tables S9 to S12. In order to alleviate the text, the Supplementary Material is not always directly referred to in this section.

First, the results for the Swiss Plateau catchments are presented and discussed, followed by the results for the Alpine catchments. While water temperature is the main focus, relevant findings regarding simulated discharge changes in the future are also presented. In the discussion, results are compared to trends and behaviours observed in past periods and related previous
publications. When comparing to data of past periods, we refer to the observations described in the work of Michel et al. (2020) unless stated otherwise.

### 5.1 Swiss Plateau Catchments

The model results for CC simulations over the Plateau catchments are similar among all considered catchments. Figure 5 shows the combined results for all considered Plateau catchments. Boxplots in the figures in this section are constructed from
all climate change scenarios and all individual years, so the range shows both the model uncertainty, the natural inter-annual variability, and catchments variability when multiple catchments are combined. The similarities in water warming between catchments show that catchment size does not play a role for the warming rate, as already observed for past periods.

For short-term projections, i.e. the period 2030-2040, the mean trend of averaged annual water temperature for the Plateau catchments is +0.26 ± 0.07 °C per decade (combining all 3 RCPs, the uncertainty indicated is the standard deviation), which is
in line with the +0.33 ± 0.03 °C per decade observed over entire Switzerland for the period 1979-2018. No significant annual discharge trends are modelled for this period.



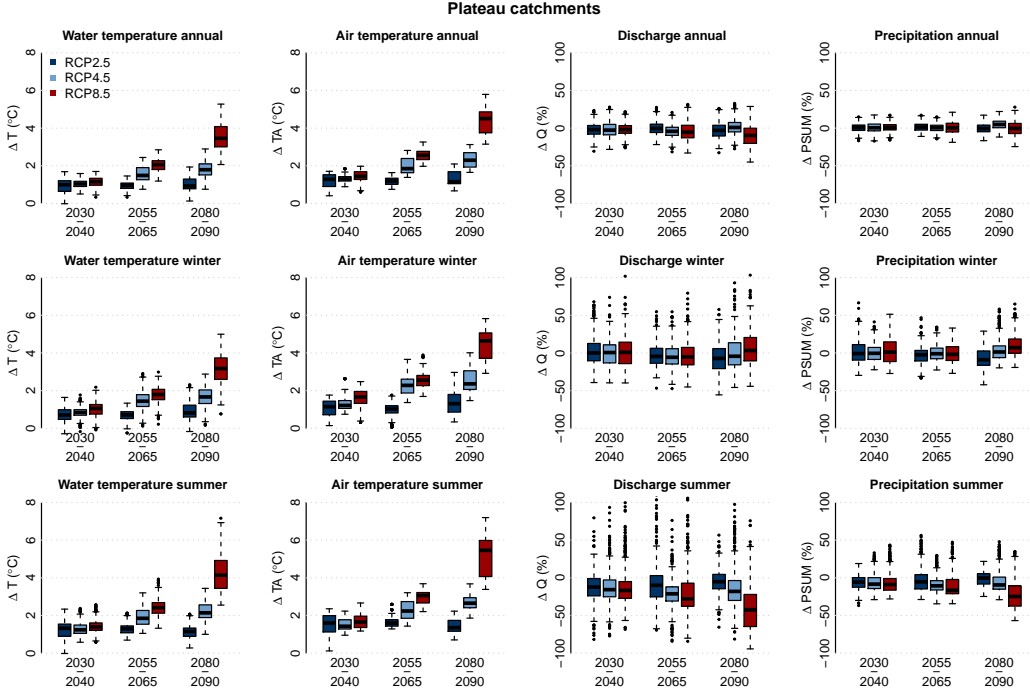

**Figure 5.** Changes in water temperature ($\Delta$T), air temperature ($\Delta$TA), discharge ($\Delta$Q), and precipitation ($\Delta$PSUM, from left to right column) over the periods 2030-2040, 2055-2065, and 2080-2090, compared to the reference period 1990-2000 for the Swiss Plateau catchments and for the 3 RCPs. The first row shows the annual changes, the second row the winter seasonal changes, and the last row the summer seasonal changes.

Over the same period, the mean air temperature trend over Swiss Plateau catchments is $0.32 \pm 0.09$ °C per decade, corresponding to a ratio between water and air temperature trends of 0.8 for this period, which compares very well to the ratio observed from historical observations. This result underlines the ability of the model chain to correctly capture the observed changes in the contemporary period. The expected warming is consistently more pronounced in summer ($+0.32 \pm 0.11$ °C per

5   decade) than in winter ($+0.21 \pm 0.9$ °C per decade) for all studied catchments.

For the periods 2055-2065 and 2080-2090, some differences between the RCP emission scenarios appear, due to the growing difference between the forcing climate scenarios. For RCP2.6, no relevant additional changes are expected beyond 2030-2040. For RCP4.5, the situation between 2055-2065 and 2080-2090 remains similar, while for RCP8.5 there is an acceleration of the changes in discharge and temperature. For the period 2055-2065, the median annual increase ranges from +1.0°C for RCP2.6

10   to +2.1°C for RCP8.5 in absolute values compared to the reference period, and some specific summers in RCP8.5 are close to +4°C. By the end of the century, the median annual water temperature increase reaches +3.5°C for RCP8.5.

These results are in line with recent predictions of Swiss lake surface water temperature over the 21st century found by Råman Vinnå et al. (2021). The increase in river temperature at the end of the century with RCP8.5 ranges between +3.2 and





**Table 5.** Change of annual mean water temperature for all 3 periods and 3 RCPs compared to the reference period 1990-2000. The median value of all years and scenarios is indicated along with the range of the values.

| Catchment | △ Water temperature (°C) | | | | | | | | |
| | | 2030-2040 | | | 2055-2065 | | | 2080-2090 | |
| | RCP2.6 | RCP4.5 | RCP 8.5 | RCP2.6 | RCP4.5 | RCP 8.5 | RCP2.6 | RCP4.5 | RCP 8.5 |
|---|---|---|---|---|---|---|---|---|---|
| **Plateau** | +0.9 [+0.1,+1.4] | +1.1 [+0.5,+1.4] | +1.1 [+0.5,+1.5] | +0.9 [+0.3,+1.2] | +1.5 [+0.8,+2.1] | +2.1 [+1.2,+2.5] | +0.9 [+0.1,+1.6] | +1.7 [+0.8,+2.5] | +3.3 [+2.1,+4.4] |
| Birs | +1.0 [+0.2,+1.3] | +1.0 [+0.8,+1.3] | +1.2 [+0.5,+1.5] | +1.0 [+0.5,+1.2] | +1.4 [+1.0,+2.1] | +2.0 [+1.4,+2.4] | +0.9 [+0.4,+1.5] | +1.8 [+1.2,+2.4] | +3.3 [+2.5,+4.4] |
| Broye | +1.1 [+0.0,+1.6] | +1.1 [+0.7,+1.4] | +1.2 [+0.5,+1.7] | +1.0 [+0.5,+1.3] | +1.5 [+0.9,+2.3] | +2.2 [+1.4,+2.6] | +1.0 [+0.3,+1.7] | +1.8 [+1.1,+2.5] | +3.6 [+2.6,+4.7] |
| Ergolz | +1.0 [+0.2,+1.3] | +1.0 [+0.7,+1.2] | +1.1 [+0.5,+1.4] | +0.9 [+0.5,+1.2] | +1.3 [+0.9,+2.1] | +1.9 [+1.4,+2.4] | +0.9 [+0.4,+1.5] | +1.7 [+1.1,+2.3] | +3.2 [+2.3,+4.4] |
| Eulach | +1.0 [-0.0,+1.4] | +1.0 [+0.6,+1.2] | +1.0 [+0.3,+1.6] | +1.0 [+0.4,+1.3] | +1.4 [+0.8,+2.3] | +1.9 [+1.2,+2.6] | +0.9 [+0.3,+1.6] | +1.6 [+1.0,+2.4] | +3.2 [+2.4,+4.6] |
| Kleine Emme | +1.1 [+0.3,+1.7] | +1.2 [+0.7,+1.5] | +1.3 [+0.7,+1.7] | +1.0 [+0.7,+1.5] | +1.7 [+1.0,+2.4] | +2.4 [+1.6,+2.8] | +1.1 [+0.4,+1.9] | +2.0 [+1.3,+2.9] | +4.2 [+2.8,+5.3] |
| RHB | +1.0 [+0.3,+1.5] | +1.1 [+0.5,+1.6] | +1.1 [+0.5,+1.6] | +1.0 [+0.5,+1.4] | +1.4 [+0.8,+2.4] | +2.1 [+1.4,+2.7] | +0.9 [+0.3,+1.8] | +1.7 [+1.0,+2.6] | +3.6 [+2.2,+5.0] |
| Suze | +0.9 [+0.1,+1.4] | +1.1 [+0.5,+1.4] | +1.1 [+0.5,+1.5] | +0.9 [+0.3,+1.2] | +1.5 [+0.8,+2.1] | +2.1 [+1.2,+2.5] | +0.9 [+0.1,+1.6] | +1.7 [+0.8,+2.5] | +3.3 [+2.1,+4.4] |
| **Alpine** | +0.7 [+0.3,+1.2] | +0.8 [+0.6,+1.1] | +0.9 [+0.4,+1.4] | +0.8 [+0.5,+1.2] | +1.3 [+0.7,+2.0] | +1.7 [+1.1,+2.3] | +0.8 [+0.4,+1.6] | +1.6 [+0.9,+2.4] | +3.3 [+1.9,+4.7] |
| Inn | +0.8 [+0.3,+1.3] | +0.9 [+0.5,+1.3] | +0.8 [+0.6,+1.4] | +0.8 [+0.5,+1.2] | +1.3 [+0.7,+2.1] | +1.8 [+0.9,+2.6] | +0.8 [+0.2,+1.3] | +1.5 [+0.5,+2.3] | +3.1 [+2.1,+4.6] |
| Kander | +0.6 [+0.3,+0.9] | +0.7 [+0.3,+0.9] | +0.7 [+0.4,+1.1] | +0.7 [+0.3,+1.2] | +1.2 [+0.5,+1.8] | +1.6 [+0.8,+2.5] | +0.7 [+0.3,+1.4] | +1.4 [+0.7,+2.4] | +2.9 [+1.4,+4.3] |
| Landwasser | +1.0 [+0.3,+1.6] | +1.1 [+0.6,+1.4] | +1.2 [+0.6,+1.7] | +1.0 [+0.6,+1.5] | +1.7 [+0.5,+2.4] | +2.2 [+1.1,+3.1] | +1.1 [+0.2,+1.8] | +1.9 [+0.9,+2.8] | +3.9 [+2.3,+5.3] |
| Lonza | +0.8 [+0.1,+1.2] | +0.8 [+0.5,+1.1] | +0.9 [+0.6,+1.3] | +0.8 [+0.4,+1.3] | +1.3 [+0.4,+1.9] | +1.8 [+0.8,+2.6] | +0.9 [-0.0,+1.4] | +1.5 [+0.8,+2.5] | +3.1 [+1.6,+4.8] |
| Lutschine | +0.7 [+0.3,+1.2] | +0.8 [+0.6,+1.1] | +0.9 [+0.4,+1.4] | +0.8 [+0.5,+1.2] | +1.3 [+0.7,+2.0] | +1.7 [+1.1,+2.3] | +0.8 [+0.4,+1.6] | +1.6 [+0.9,+2.4] | +3.3 [+1.9,+4.7] |

+3.3°C in all seasons, except in summer when an increase of +4.1°C is predicted. For some summers and CC scenarios, the warming can reach up to +7°C. Changes in annual discharge patterns, linked to precipitation changes, appear with RCP4.5 and RCP8.5 for the periods 2055-2065 and 2080-2090 (more marked for RCP8.5 and for the latter period). An increase in winter discharge and a decrease in summer discharge are expected, with no significant change at annual scale.

5     Historical data do not exhibit a strong correlation between discharge and water temperature in summer, except in warm and dry summers when low discharge exacerbates the warming. In central Europe, a clear link between dry spells and heat waves in summer has been found (Fischer et al., 2007b, a) and they both are expected to increase in the future (MeteoSuisse et al., 2018a). However, as explained by Michel et al. (2021b), the methodology used for the temporal downscaling will "smooth-out" the information about this kind of extreme events. As a consequence, these events are not present in the simulation performed





here, and it is likely that more important warming than predicted here will arise during extremely warm and dry summers in the future.

Despite these limitations, the interaction between discharge and water temperature can still be investigated with the model results, but again, due to the time series used, only at seasonal scale. The correlations between changes in water temperature

($\Delta$T), air temperature ($\Delta$TA), and discharge ($\Delta$Q) in summer and winter are shown in Table 6. A correlation between $\Delta$T and $\Delta$TA is present in both seasons. The absence of correlation between $\Delta$TA and $\Delta$Q in winter suggests that snowmelt only plays a minor role for Plateau catchments. In summer, there is a negative correlation between $\Delta$T and $\Delta$Q; but at the same time, $\Delta$Q is also strongly negatively correlated with $\Delta$TA, meaning that we cannot show here any strong direct impact of $\Delta$Q on $\Delta$T in summer. Additionally, a plot of $\Delta$T versus $\Delta$Q in summer (Figure 6) does not suggest any noticeable impact of change

in discharge on change in water temperature at seasonal scale during summer on the Swiss Plateau, even for extremely warm summers, and the main forcing variable for water temperature clearly is the air temperature. Further investigation of the Q-T relation under climate change, especially at short time scales, would be highly desirable but cannot be achieved with the CC scenarios used in this work.

Nevertheless, the significant decrease in summer discharge predicted by the end of the century (median -5% for RCP2.6,

-18% for RCP4.5 and -42% for RCP8.5, all three with considerable variability) will have some important consequences in terms of water availability and will impact future management of water resources. For the Broye catchment, the decrease in discharge reaches a median value of -75% in summer for RCP8.5 at the end of the century. The observed decrease in summer discharge is always larger than the decrease in precipitation from CC models due to the increase of the evapotranspiration (ET) fraction. Simulated changes in ET for four catchments are show in Figure 7. Results show an increase in ET of about +15%

by the end of the century, combined with a deficit of precipitation of about -25% for the emission scenario RCP8.5. A special case is the Eulach catchment, which is much more urbanised compared to the other studied catchments, explaining the lower change in ET observed there. The changes in ET are similar for the RCP4.5 and RCP8.5 scenarios and only about 1.5 times larger than the one expected with the RCP2.6 scenarios, showing that ET will be limited mainly by water availability and that potential additional water during wetter summers will have a very high potential to evaporate. This is also clear from the large

variability in ET for the RCP8.5 scenarios.

The expected increase in water temperature will have a large impact on both natural and human systems. To evaluate this impact in the future, we use two indicators introduced by Michel et al. (2020): A) The number of days per year when the daily maximum temperature is above 25°C, which is a legal threshold in Switzerland for unrestricted water usage, i.e. for industrial cooling, and B) a metric indicating the number of days per year when salmonid fish are exposed to Proliferative Kidney Disease

(PKD). The latter metric is based on the model developed by Carraro et al. (2016) counting the number of days per year for which the minimal daily temperature exceeds 15°C for at least 28 consecutive days. Values for these two indicators are shown for 4 catchments in Figures 8 and 9. Similar values based on measured water temperature have been computed using past measurements for the period 1990-2000 for the Broye and the Birs (only the second indicator for the latter).

The values obtained over the historical period here match very well the values obtained from measurements in the work of

Michel et al. (2020), showing again the robustness of the results obtained when the models are forced with CC scenarios. For



**Table 6.** Correlation between changes in water temperature ($\Delta$T), air temperature ($\Delta$TA), and discharge ($\Delta$Q) between the period 1990-2000 and 2080-2090 for RCP8.5 scenarios for all considered Swiss Plateau catchments (top part) and Alpine catchments (bottom part). The correlation is computed with the Pearson correlation and only significant results (p-value<0.05) are shown.

| | **Swiss Plateau catchments** | | |
|---|---|---|---|
| **Season** | **Cor $\Delta$T $\Delta$TA** | **Cor $\Delta$Q $\Delta$TA** | **Cor $\Delta$T $\Delta$Q** |
| DJF | 0.62 | - | - |
| JJA | 0.85 | -0.50 | -0.60 |
| | **Alpine catchments** | | |
| **Season** | **Cor $\Delta$T $\Delta$TA** | **Cor $\Delta$Q $\Delta$TA** | **Cor $\Delta$T $\Delta$Q** |
| DJF | 0.30 | 0.14 | 0.60 |
| JJA | 0.85 | -0.37 | -0.52 |

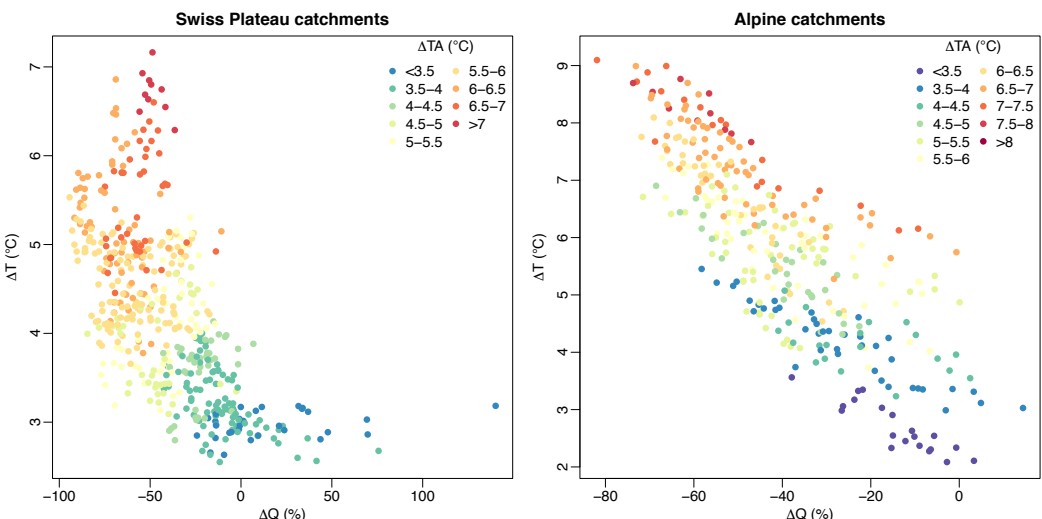

**Figure 6.** Changes in water temperature ($\Delta$T) versus changes in discharge ($\Delta$Q) during the summer season for the period 2080-2090 compared to the reference period 1990-2000 for RCP8.5 emission scenarios and for all considered Swiss Plateau catchments (left) and Alpine catchments (right). Colors indicate the corresponding change in air temperature ($\Delta$TA).

each of the 4 catchments in Figures 8 and 9, both indicators grow over time and with increased greenhouse gas emissions. Catchments such as the Birs and the Eulach, which are currently less subject to high water temperature, will reach both the legal threshold of 25°C and the PKD exposure limit rather often in the future. By the middle of the century PKD will find favourable conditions to spread in summer in all 7 investigated catchments (for all RCPs), with possibly devastating impact on

5    salmonid fish population. For catchments with relatively warm water temperatures in current conditions, such as the Broye and the Kleine Emme, the legal limit of 25°C will be reached almost every year already by 2030-2040 regardless of the emission




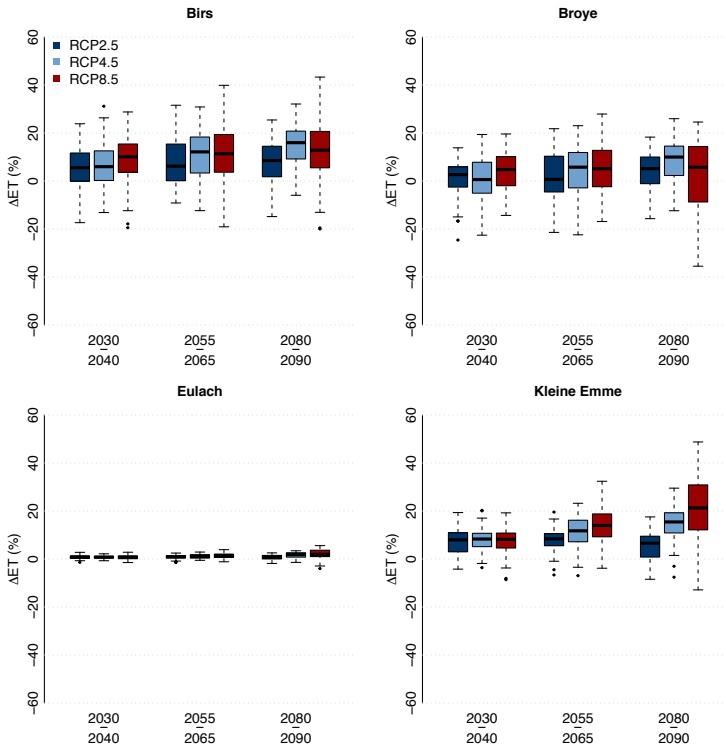

**Figure 7.** Relative changes in evapo-transpiration (ΔET) over the periods 2030-2040, 2055-2065, and 2080-2090, compared to the reference period 1990-2000 for 3 RCPs and for 4 catchments of the Swiss Plateau region. These 4 catchments were chosen to represent the variation in Swiss Plateau catchment type.

scenarios. This will persist for future time periods, even under RCP2.6, which already can be considered a target with low probability (Stocker, 2013). By the end of the century and with high emission scenarios, the water temperature will be above this threshold for around 2 months per year in these two catchments. This will entail either stopping regular water usage for industry and cooling in such catchments, or a necessary adaptation of current regulation, at the risk of further enhancing the impacts and increasing the stress and pressure on these ecological systems.

## 5.2 Alpine catchments

### 5.2.1 Overview

At annual scale, the water temperature rise observed in Alpine catchments is close to that observed over the Swiss Plateau (see Table 5), despite a slightly higher air temperature increase (see Figures 5 and 10). Warming is more limited for the early periods in Alpine environments, but it reaches the same level as the Plateau catchments toward the end of the century. This



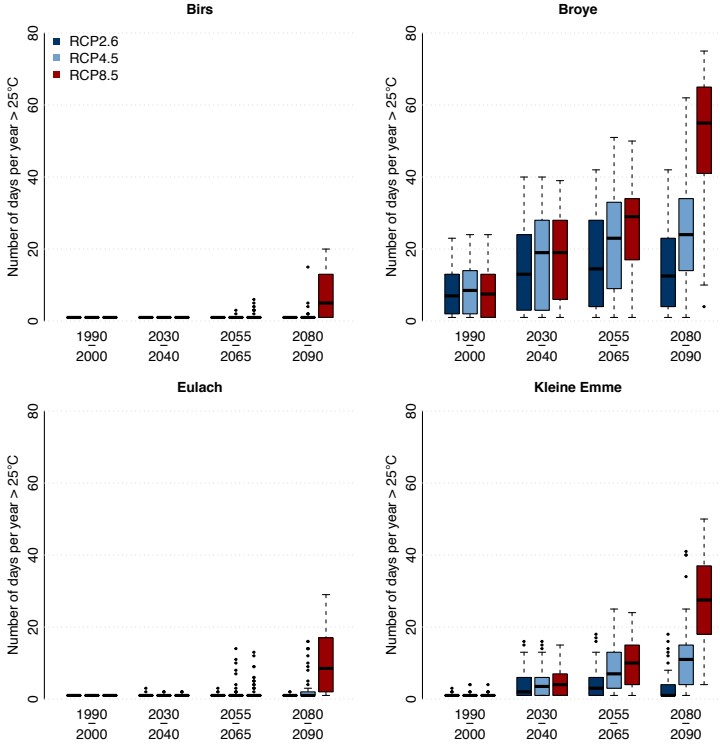

**Figure 8.** Number of days per year when the temperature is above the threshold of 25°C, for each year in each time period and for each scenario, for 4 Swiss Plateau catchments. These four catchments were chosen to represent the variation in the Swiss Plateau catchment type.

slower warming in Alpine catchments compared to Swiss Plateau catchments in early periods of the 21st century is coherent with current trends observed in Switzerland.

At seasonal time scale, in turn, water temperature change is very different between Swiss Plateau and Alpine catchments. While in the Plateau catchments the warming is similar in all seasons except for summer, the seasonal pattern is completely
5 different in Alpine catchments, as demonstrated in Figure 10, cf. Figure 5 for comparison. Furthermore, to expand the analysis, Figure 11 shows the evolution of snow water equivalent, of solid precipitation and of soil surface temperature on yearly and seasonal basis, for all considered Alpine catchments combined (for individual catchments, see Figures S62 to S67). The evolution of the annual discharge and temperature cycles for the 3 time periods and 3 RCPs is exemplified for the Inn catchment in Figure 12 (see also S68 to S71 for the other Alpine catchments).

10 **5.2.2 Winter**

During the winter season, despite an air temperature increase similar to that projected for the Swiss Plateau, the water temperature increase is very limited, reaching only a median value of +1.3°C at the end of the century with RCP8.5 scenarios. This reduced winter warming in Alpine catchments is consistent with observations from the past decades. At the same time,





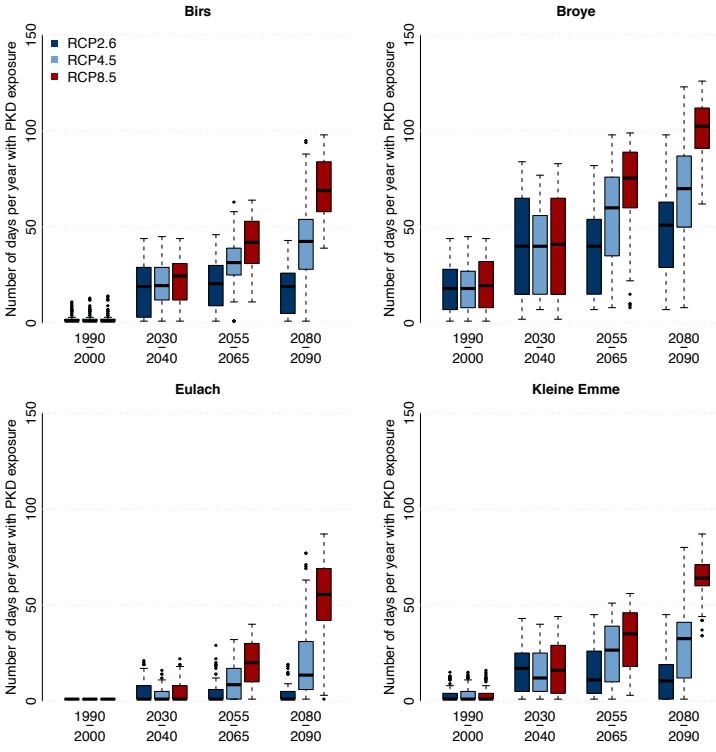

**Figure 9.** Number of days per year when salmonid populations are exposed to PKD, based on the metric presented in the work of Michel et al. (2020), for each year in each time period and for each scenario, for 4 Swiss Plateau catchments. These four catchments were chosen to represent the variation in the Swiss Plateau catchment type.

an increase in discharge between +10% in 2030-2040 (for the 3 RCP scenarios) and up to +40% for RCP8.5 at the end of the century is expected. Except for RCP8.5 at the end of the century, no significant difference in overall precipitation is expected. However, a decrease is expected in solid precipitation in winter, along with a significant decrease in mean SWE of up to -50% for RCP8.5 at the end of the century (Figure 11). The fact that the decrease in mean SWE is more important than the decrease

5    in solid precipitation shows that more melt is expected during the winter season in the future. The combination of the lower fraction of solid precipitation (i.e. more rain) and the enhanced snowmelt explains the increase in winter discharge.

A weak but significant positive correlation between $\Delta T$ and $\Delta Q$ in winter is found (see Table 6), analog to results from past observations. This is partly explained by the decreased sensitivity of water temperature to air temperature when more water (mass) is present.

10    The most important limiting factor for the Alpine river temperature rise in winter (even with a mean air temperature rise up to +4.3°C for RCP8.5 at the end of the century compared to the reference period), is simply that the air temperature mostly remains below freezing at higher elevations, especially during night. In these periods, the water temperature stays above the

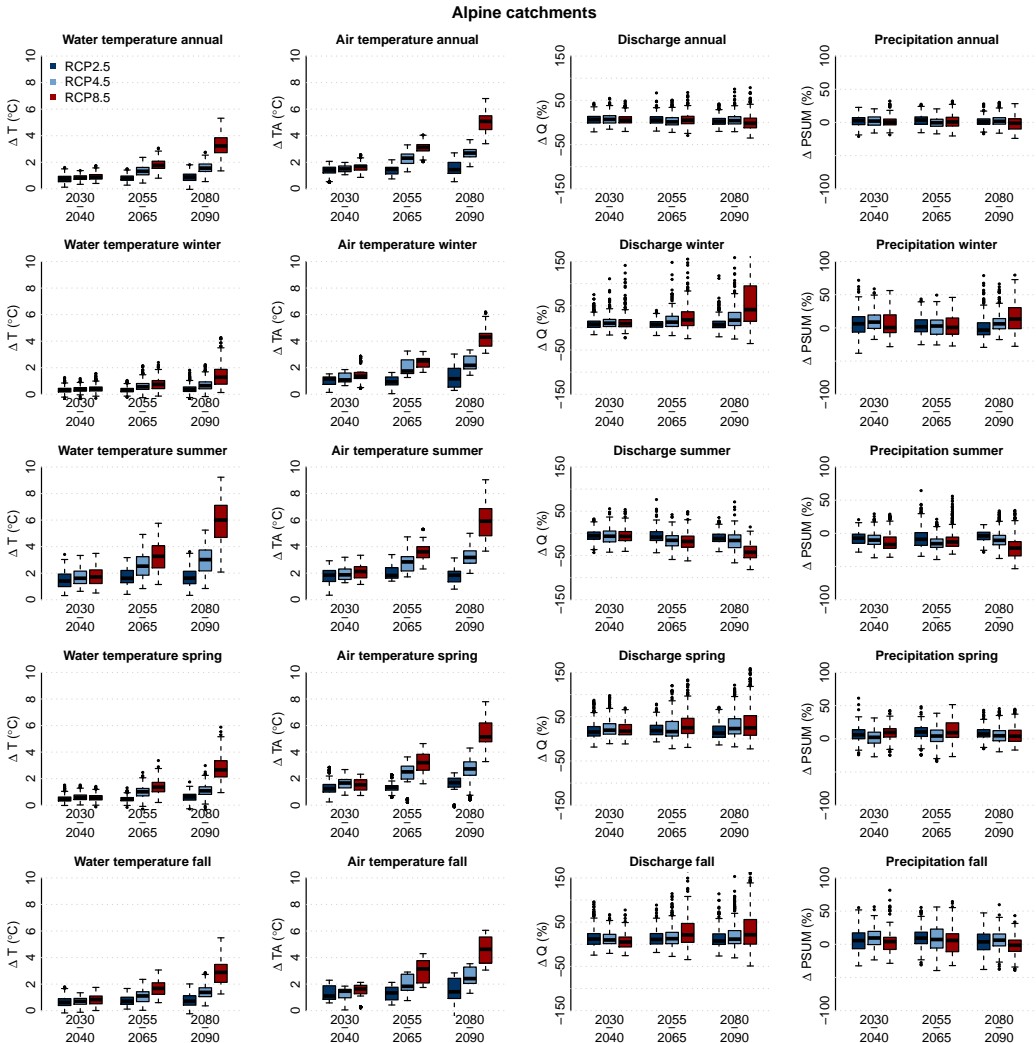

**Figure 10.** Changes in water temperature ($\Delta T$), air temperature ($\Delta TA$), discharge ($\Delta Q$), and precipitation ($\Delta PSUM$, from left to right column) over the periods 2030-2040, 2055-2065, and 2080-2090 and for the 3 RCPs, compared to the reference period 1990-2000, averaged over all considered Alpine catchments. Row 1 shows the annual change, row 2 the winter seasonal change, row 3 the summer seasonal change, row 4 the spring seasonal change, and row 5 the fall seasonal change.

air temperature and does not experience any warming. In addition, for near-future periods or low emission scenarios, the snow cover often prevents an increase in soil temperature in winter (Figure 11).



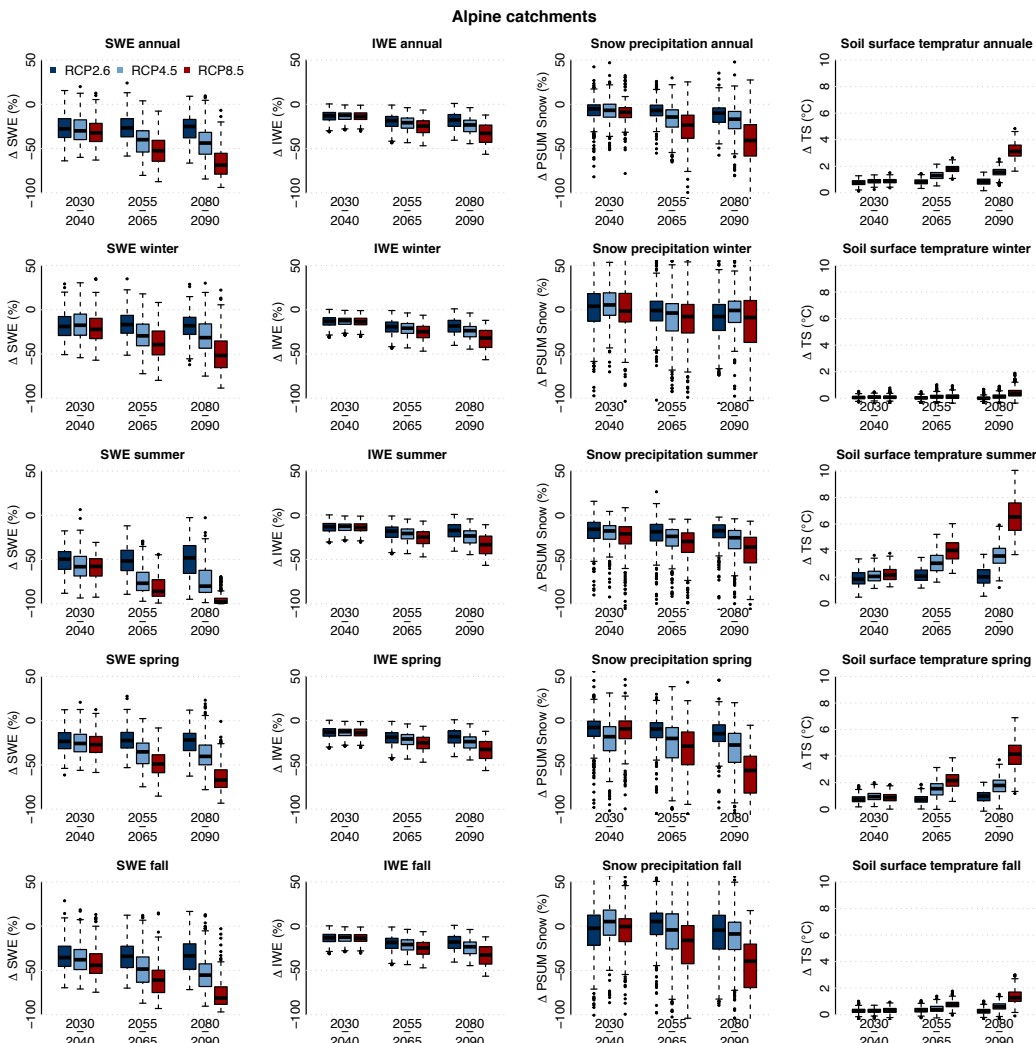

**Figure 11.** Changse in snow water equivalent in the catchments (ΔSWE), in ice water equivalent (mass stored in glaciers and in snow in glacier pixels) in the catchments (ΔIWE), in solid precipitation (ΔPSUM snow), and in soil surface temperature (ΔTS, from left to right column) over the periods 2030-2040, 2055-2065, and 2080-2090 and for the 3 RCP scenarios, compared to the reference period 1990-2000, averaged over all considered Alpine catchments. Row 1 shows the annual change, row 2 the winter seasonal change, row 3 the summer seasonal change, row 4 the spring seasonal change, and row 5 the fall seasonal change.

### 5.2.3 Spring

During the spring season, a shift toward earlier snowmelt is illustrated by a substantial decrease in SWE (Figure 11), which under RCP8.5 is already more marked in the period 2045-2055 and gets even more pronounced toward the end of the century. Note that for computing correct snowmelt delta in spring, the winter change in SWE has to be subtracted from the change

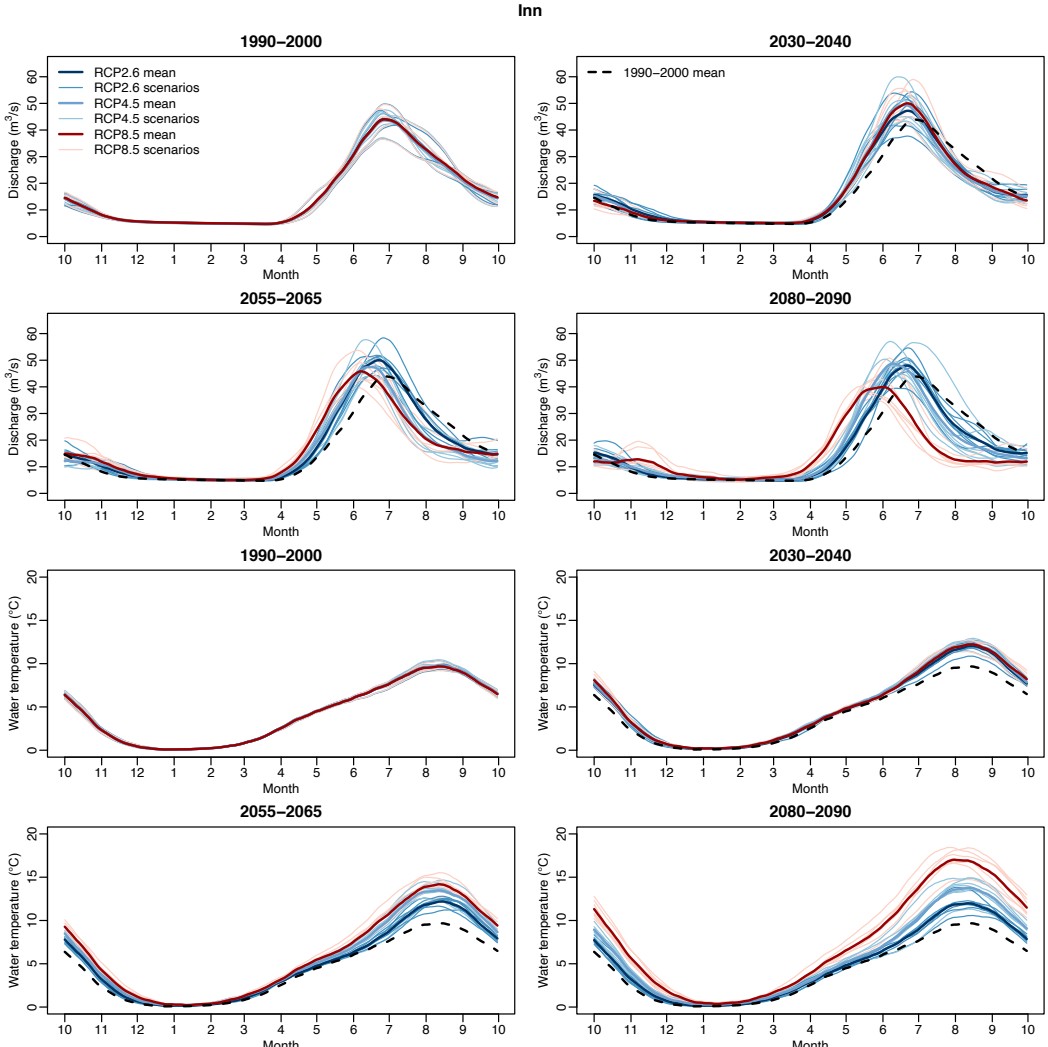

**Figure 12.** Annual cycle of discharge (top) and temperature (bottom) for the Inn catchment. The cycles are obtained by computing the average for each day of the year and by applying a circular moving average of 30 days. Dark lines show the mean for each RCP over each period, light lines show individual scenarios. Black dashed lines indicate the mean over the reference period 1990-2000 (only shown in subsequent periods to ease comparison).

observed in spring. Indeed, future reduced SWE in spring is not only due to enhanced melt, but also a result from a lower initial SWE at the beginning of the season. For RCP8.5 at the end of the century, the increase of melt in the spring season is about 20%. In addition, a significant reduction of solid precipitation is expected in future spring seasons. These effects combined lead to a considerable increase in discharge and a shift of the peak runoff toward earlier in the year. Even for the low and moderate





emission scenarios RCP2.6 and RCP4.5, this shift is clearly visible at the end of the century, and for RCP8.5, we also observe a flatter peak anticipated by almost 2 months. These results are consistent with the study on discharge by Muelchi et al. (2020).

Despite this increase in spring discharge, the river warming is more marked in spring than in winter for the periods 2055-2065 and 2080-2090, owing to a significant increase in spring air and soil temperature. While in winter the snow cover is still

present and the negative air temperatures do not lead to any significant soil surface warming, in spring, the soil surface warms almost at the same rate as the air, due to a smaller snow covered area. This soil warming effect related to a reduced spatial and temporal seasonal snow cover extent is in reality closely linked to potential cold water inputs from surface runoff or from otherwise highly connected cold water storage. However, as amply discussed in Sections 3 and 4.1.2, the advection of such cold water is not captured in the model chain used. Accordingly, the water temperature warming might be overestimated in

spring since the predicted enhanced melting might inject considerable amounts of cold water to the stream netw ork.

### 5.2.4   Summer

The summer season shows distinctively different water temperature change patterns and intensity between RCPs and time periods. For the near future (2030-2040), the expected water temperature increase remains below the air temperature increase, similar as for the Swiss Plateau catchments. Advancing in time and looking especially at RCP8.5 scenarios, the water tempera-

ture warming catches up with the air temperature warming and even slightly exceeds it during the warmest summers, leading to a median water temperature warming of about $+6.0°C$ ($+5.9°C$ for the air temperature). Some summers in some CC scenarios can exhibit a water temperature warming of $+9°C$. Again, the discussion in Sections 3 and 4.1.2 on the errors observed during model calibration for summer over the Alpine catchments requires a critical analysis of these values.

In summer, the median of the SWE decrease for RCP8.5 by the end of the century is -100%, meaning that all remaining snow

is melting in the majority of the scenarios and of the years early in the summer. However, the SWE decrease in spring, about -70%, means that the available snow to be melted in summer is reduced by -70%. The glacier melt will also be less important in the future. The cold water advection resulting from snowmelt, not represented in the model, will thus be relatively less important in the future than in current conditions. It will be rather important in spring as discussed above. The large decrease in summer snow cover and the retreat of glaciers, causing a reduction of surface albedo and the loss of the insulating snow layer,

lead in these areas to a drastic soil surface warming. In 2030-2040 (all scenarios) and during the second part of 21st century under RCP2.6, this warming remains limited, with increases below those of air temperature. However, for RCP4.5 and RCP8.5 and periods further ahead, these changes in land cover cause an important soil temperature warming, which can exceed the warming of the air temperature, and which contributes thereby to the substantial increase in water temperature.

The discharge decrease also plays a role in the simulated warming in summer. For the Plateau catchments, the positive

correlation between summer $\Delta T$ and $\Delta Q$ (Table 6) was shown to be rather driven by the underlying correlation of both variables with air temperature. This is different in the Alpine catchments. Despite similar correlation values in Table 6, plot of $\Delta T$ versus $\Delta Q$ in Alpine catchments shows a different picture than in the Swiss Plateau (Figure 6). That is, for a similar range of air temperature increase, the water temperature increase aligns well with the discharge decrease. Such relationship is mostly absent in the Swiss Plateau catchments.





### 5.2.5 Fall

During fall, a discharge reduction occurs at the beginning of the season, caused by the shift of annual peak discharge to earlier in spring and summer, followed by a discharge increase later in the season due to an increased fraction of liquid precipitation and rapid melting of occasional snowfall. These fall melt events contribute to cool the soil; accordingly, the overall soil warming

in fall remains limited and significantly below the prediction for the Swiss Plateau catchments.

At the end of fall, and similar to winter, air temperature is mostly negative, limiting thereby the warming of water and leading to limited water temperature increase over fall, on average. However, as depicted in Figure 12, the future water temperature in September will be much higher than during past decades, meaning that the period when the ecosystem will be affected by an important warming will extend from June to the end of September.

**5.3   Role of discharge variations for summer water temperature**

To further investigate the relationship between water temperature and discharge during summer, uni- and multivariate linear models on $\Delta$T are used, taking $\Delta$TA, $\Delta$Q, or both as predictors. The models are applied separately for the Swiss Plateau and for the Alpine catchments for the summer season, the period 2080-2090, and RCP8.5 (Table 7). For the Swiss Plateau catchments, both $\Delta$TA and $\Delta$Q are significant predictors when used separately (with a lower predictive power when using $\Delta$Q), but when

used together the significance of $\Delta$Q disappears and the explanatory power of the model is not improved compared to using $\Delta$TA alone. This confirms the hypothesis that changes in discharge have no impact on water temperature change. In Alpine catchments, however, $\Delta$Q remains significant in the multivariate model and increases the $R^2$ value from 0.73 (when using only $\Delta$TA) to 0.89. In addition, using $\Delta$Q alone allows to explain 2/3 of the variability in $\Delta$T. This analysis shows that the change in discharge influences the change in water temperature in Alpine catchments during summer, while it has no effect on Swiss

Plateau catchments.

There is no straightforward explanation for this different sensitivity of water temperature change to discharge change between Swiss Plateau and Alpine catchments. The most likely reason is the difference in flow regime between Plateau and Alpine rivers. In Swiss Plateau catchments, the summer discharge is already low nowadays compared to the other seasons, meaning that a decrease in summer discharge will not necessarily increase the sensitivity to air temperature. In Alpine regions,

the annual cycle is more pronounced and the summer season is currently characterized by high discharge. The expected shift in peak discharge from mid-summer to late spring and the low flow conditions at the end of the summer expected in the future might lead to an increased sensitivity to air temperature. This growing sensitivity of water temperature, amplified by discharge reduction in summer, contributes to the different summer warming intensity in the Alpine catchments compared to the Swiss Plateau catchments. Note that for the early period 2030-2040 or with low emission scenarios (i.e. in the case of a moderate

impact of CC on snow and glacier cover and of a less radical shift in discharge regime), the increase of water temperature in summer in Alpine catchments is similar to the increase simulated in Swiss Plateau catchments. The differences appear only later and with high emission scenarios along with important changes in the snow and glacier cover and of the discharge regime.





This suggests that a potentially overestimated warming simulated in Alpine catchments during the calibration and validation periods does not affect the estimation of future warming to a large extent.

**Table 7.** Summary of the linear models for changes in water temperature ($\Delta T$) using changes in air temperature ($\Delta TA$), water discharge ($\Delta Q$), or both, as predictors. Changes in the period 2080-2090 compared to the reference period 1990-2000 are used for RCP8.5. The table shows the coefficients, the *p-values* associated to each predictor, and the adjusted $R^2$ (discounting the effect of additional explanatory variables) of each model. The linear models are applied separately for the Swiss Plateau catchments (top) and the Alpine catchments (bottom).

| Swiss Plateau catchments | | | |
|---|---|---|---|
| **Predictor(s)** | **Coefficient(s)** | ***p-value(s)*** | $R^2$ |
| $\Delta TA$ | $0.76 \pm 0.02$ | <2e-16 | 0.74 |
| $\Delta Q$ | $0.020 \pm 0.001$ | <2e-16 | 0.38 |
| $\Delta TA$ and $\Delta Q$ | $0.77 \pm 0.03$ and 6e-4 $\pm$ 1e-3 | <2e-16 and 0.58 | 0.74 |
| **Alpine catchments** | | | |
| **Predictor(s)** | **Coefficient(s)** | ***p-value(s)*** | $R^2$ |
| $\Delta TA$ | $1.07 \pm 0.03$ | <2e-16 | 0.73 |
| $\Delta Q$ | $-0.071 \pm 0.002$ | <2e-16 | 0.67 |
| $\Delta TA$ and $\Delta Q$ | $0.72 \pm 0.03$ and $-0.042 \pm 0.002$ | <2e-16 and <2e-16 | 0.89 |

## 5.4 Robustness, limitations and open questions

Over the Swiss Plateau, the results obtained for the discharge agree well with a previous CC study for Switzerland (Muelchi
et al., 2020), and the median annual temperature increase modeled is similar to what is expected in the future for Swiss lakes (Råman Vinnå et al., 2021). The few studied catchments can be assumed to be representative of Swiss catchments in general (Michel et al., 2020), and the results obtained here can be considered as representative baseline of future trends.

Overall, the model shows good performance during the validation period. The errors in water temperature (RMSE) observed during the calibration and validation periods are far below the CC signal for RCP4.5 and RCP8.5, which underlines the robust-
10 ness of the simulated trends. In Alpine catchments, despite of some discrepancies between modelled and observed summer temperatures, we argue that the CC results can also be considered as being robust, as summarized below.

1. The proposed cause for the overestimation of summer water temperatures (cold water input parameterization) has a relatively minor impact when using CC scenarios because cold water input is reduced for all scenarios and future time periods.

2. The results obtained for the nearby future (2030-2040) or with low emission scenarios are in agreement with the observations performed over the last four decades in Switzerland (Michel et al., 2020) (both for the different behaviours observed in Swiss Plateau and Alpine catchments).



3. The simulations show stronger CC impacts on water temperature in medium or high emissions scenarios and towards the end of the century, especially in Alpine catchments. In the simulations, this more important impact in Alpine catchments is explained by an increase in soil temperature and a reduction in discharge, jointly leading to higher sensitivity of water to air temperature in summer. These simulated processes are also in line with observations from recent warm years in Alpine catchments with reduced snow cover (Michel et al., 2020).

In essence, despite the higher complexity in modelling the Alpine regions compared to the lower elevations, we have shown strong evidence that the simulated differential warming between Alpine and Swiss Plateau catchments is consistent with observations, robust and plausible. Nevertheless, we interpret the results on Alpine catchments as first results of CC simulations in this type of catchments, providing important new information and insights, but also with several sources of uncertainty.

There are a number of shortcomings in the used models. In addition to the lack of parameterization of cold water input to the stream network, no dynamic interaction with the water table is taken into account. Further model improvement to better represent the interplay of soil infiltration, cold water advection and groundwater dynamics might be of key importance for future CC impact assessments on water temperature because this might come in parallel with newly developed methods for groundwater and irrigation water management (García-Gil et al., 2015), which might even include specific water heat management, such as winter water infiltration for summer river cooling (Epting et al., 2013).

This also relates to another crucial limitation of the setup used: the fact that only mostly natural undisturbed river courses are studied since the model does not deal with any kind of anthropogenic structures, such as dams, pumping, deviations, intakes or rejections, which have important impacts on water temperature (Michel et al., 2020; Seyedhashemi et al., 2021). The patterns of water usage for irrigation, electricity production or industrial cooling will likely evolve and change in the future due to CC.

In addition, Michel et al. (2020) showed with historical data that the presence of lakes along the watercourse changes the warming rate of rivers. This two-way interaction in river-lake-river systems also remains to be investigated in more detail in future works, especially taking into account the important shifts in the Alpine flow regime simulated here and in lake mixing (Råman Vinnå et al., 2021).

It is also noteworthy that extreme events are not covered here because the used models are not validated for extreme events and the forcing time series are not suited. It thus remains to be analysed how future heat waves and dry spells will influence water temperatures in Switzerland.

Finally, both the downscaling method used for climate change scenarios and the computational limitations forbid transient runs over the whole 21st century. Transient runs would be informative about the time of emergence of transition in water temperature such as the ones predicted for Alpine catchments.

# 6 Conclusion

This work presents the fist extensive study of climate change (CC) impact on rivers water temperature in Switzerland and, to the best of our knowledge, in Alpine areas. A chain of physics-based models has been used with 24 CC scenarios, spanning three different emissions pathways (RCP2.6, RCP4.5, and RCP8.5), and applied to two categories of catchments, namely lowland





Swiss Plateau catchments and high elevation Alpine catchments. The work presented here required substantial optimization work in the source codes of the models, which underlines the importance of good documentation, maintenance, accessibility, and collaboration around model source codes, which is often undervalued.

We demonstrate the ability of the developed model chain to reliably simulate the water temperature of a variety of catchments over Switzerland. The results obtained for water temperature and discharge for the near future with 21 CC scenarios are coherent with past and current observations in Switzerland and in central Europe (Moatar and Gailhard, 2006; Webb and Nobilis, 2007; Arora et al., 2016), and with other modelling studies using the same forcing scenarios over Switzerland.

Based on our modelling results, expected CC impacts on river water temperature in Switzerland can be summarized as follows:

- A clear warming of river water is modelled during the 21st century, more pronounced for the high emission RCP8.5 scenarios and toward the end of the century. For the period 2030-2040, median warming in water temperature of +1.1°C (with a range of 0.0-1.7°C for individual years, rivers and CC scenarios) for Swiss Plateau catchments and of +0.8°C (0.1°C-1.7°C) for Alpine catchments are expected compared to the reference period 1990-2000 (similar for all 3 RCP scenarios). At the end of the century (2080-2090), the median annual water temperature increase in Swiss Plateau catchments amounts to +0.9°C for RCP2.6, (with a range of 0.1-1.9°C), and to +3.5°C for RCP8.5 (2.1-5.3°C). In Alpine catchments, it amounts to +0.9°C for RCP2.6 (0.0-1.8°C) and to +3.2°C for RCP8.5 (1.4°C-5.3°C).

- On annual average, the river warming expected over Swiss plateau catchments account for around 75% of the air temperature increase. For Alpine catchments, where air temperature warming is slightly more pronounced than for the lowland plateau catchments, the annual water temperature increase corresponds to 50% of the air temperature warming for low emission scenarios or for the near future, and the ratio increases to 65% at then end of the century for high emissions scenarios.

- At the seasonal scale, the warming on the Swiss Plateau and in the Alpine regions exhibits different patterns. For the Swiss Plateau, the spring and fall warming is comparable to the warming in winter, while the summer warming is stronger but still moderate. A significant reduction in summer discharge will occur in Swiss Plateau catchments for high emission scenarios by the end of the century, with a median value of -42%. No significant discharge trends are expected, in any season, with low emission scenarios. In Alpine catchments, only a very limited warming is expected in winter. An important discharge increase in winter and spring is expected in these catchments due to enhanced snowmelt and to a larger fraction of precipitation falling as rain. Accordingly, the period of maximum discharge in Alpine catchments, currently occurring during mid-summer, will shift to earlier in the year by a few weeks (RCP2.6) or almost two months (RCP8.5) by the end of the century.

- In summer, the marked discharge reduction in Alpine catchments for high emission scenarios leads to an increase in sensitivity of water temperature to low discharge, which is not observed in the Swiss Plateau catchments. This difference in water temperature sensitivity to low discharge in summer is explained by the fact that Swiss Plateau catchments





are currently already experiencing low discharge conditions in summer, while Alpine catchments will experience an important shift in discharge seasonality. This result has not been demonstrated in previous studies.

  – In the near future or with low emission scenario, the summer warming in Alpine catchments is similar or slightly lower to the one observed in the Swiss Plateau catchments. By the end of the century and for high emission scenarios, the

reduced snow cover in spring and summer will lead to amplified warming of the soil. Along with the increased sensitivity of water temperature to low discharge conditions, this leads to a summertime river warming of +6.0°C (2.1-9.2°C) in Alpine catchments. This range is comparable to the air temperature warming.

The most unexpected result is the important water warming modelled in summer for Alpine regions. This remains to be confirmed in future studies, possibly with a modelling chain that includes an improved parameterization of cold water advection

and of groundwater table dynamics, which might also be of key importance for future joint assessment of CC and anthropogenic changes.

The results show that river systems in Switzerland (and likely the entire Alps and adjacent regions), will undergo substantial changes in the near future, both in terms of water temperature and water availability. Even for the most ambitious emission scenario RCP2.6, we demonstrate that the expected warming will have an impact on industrial water usage and on the aquatic

fauna, showing the urgent need of mitigation strategies for water temperature and for adaptation. For RCP8.5 emissions scenarios, the extreme conditions by the end of the century will threaten both ecosystems, ecosystem services and anthropogenic water use. The current rapid advances in water temperature modelling with different approaches (machine learning, statistical models or physics-based models) should thus cross the boundaries of purely scientific applications and be made available to a broader public for operational use in water temperature forecast and warning systems. Future development of monitoring

systems will also be of large importance for improving the understanding of water temperature processes and how this is represented in models.

*Author contributions.* The paper was written by Adrien Michel with contributions from all co-authors. Adrien Michel collected the data, further developed and run Alpine3D and StreamFlow, and preformed the analysis. Nander Wever helped the model development. Harry Zekollari performed the GloGEMflow simulations. Adrien Michel, Bettina Shaeffli, Michael Lehning, and Hendrik Huwald designed the

study. All authors gave critical feedback on the manuscript.

*Competing interests.* BS is editor at HESS. The other authors declare that they have no conflict of interest.

*Code and data availability.* All results produced throughout this paper are either publicly available on Envidat (for review time the data are available at: https://drive.switch.ch/index.php/s/pk2xm4u195JwFUy, password will be communicated to reviewers) or available upon request (due to the large size of the gridded data). The source code for MeteoIO, SNOWPACK, Alpine3D, and StreamFlow are avail-





able at https://models.slf.ch. The following versions have been used in this work: MeteoIO 3.0.0 (rev 2723), SNOWPACK 3.6.1 (rev 1878), Alpine3D 3.2.1 (rev 570), and StreamFlow 1.2.2 (rev 368). Additional pre- and post-processing tools along with the setup of the simulations and necessary data are available on Envidat (for review time the data are available at: https://drive.switch.ch/index.php/s/pk2xm4u195JwFUy, password will be communicated to reviewers). The source code of the model GloGEMflow can be obtained upon

request to the corresponding author. The downscaled climate change scenarios are also available on Envidat (https://www.envidat.ch/#/metadata/climate-change-scenarios-at-hourly-resolution). Unfortunately, we are not allowed to publicly share the historical meteorological and hydrological measurements, but they are available upon request from the mentioned data providers or via the corresponding author.

*Acknowledgements.* The work was funded by the Swiss Federal Office for the Environment (FOEN), Hydrology Division, CH-3003 Bern under grant no. 15.0003.PJ / Q102-0785. The Federal Office of Meteorology and Climatology (MeteoSwiss), the WSL Institute for Snow

and Avalanche Research (SLF), the Swiss Federal Office of Topography (Swisstopo), the Federal Office for the Environment (FOEN), the Office for Water and Waste of the Canton of Bern (AWA), the Office for Waste, Water, Energy and Air of the Canton Zurich (AWEL), and Holinger AG are acknowledged for free access to their data.

Petra Schmocker-Fackel and Fabia Hüsler (FOEN) are sincerely acknowledged for their support. We thank Jannis Epting (UniBas) for the instructive discussions about groundwater, Ionut Iosifescu (WSL) for his help regarding the data portal Envidat, Mathias Bavay (SLF)

for his help in Alpine3D model development, and Tristan Brauchli (Crealp) for helpful discussions throughout this project. Harry Zekollari acknowledges the funding received from WSL (internal innovative project), the BAFU Hydro-CH2018 project and a Marie Skłodowska-Curie Individual Fellowship (Grant 799904).

The vast majority of work was performed with performed with open and free languages and software (mainly C, C++, bash, Python, R, MeteoIO, Snowpack, Alpine3D, StreamFlow, TauDEM, GDAL, SQL, and QGIS, along with countless libraries), and the authors acknowl-

edge the open-source community for its invaluable contribution to science. The simulations were performed on the Piz Dainz supercomputer of the Swiss National Supercomputing Center (CSCS). The CSCS technical team is acknowledged for his help and support during this project.





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
