# Peer review of "Future water temperature of rivers in Switzerland under climate change investigated with physics-based models"

_Hydrology and Earth System Sciences, 2021_

## Author Comment (AC2)

Dear reviewer,

First of all, we would like to thank you for this clear and helpful review. We are very well aware that this review has been a significant time investment and therefore especially appreciate the reviewer's feedback and commitment. We provide our detailed answers and explanations below and hope that this address and clarify the reviewer's comments and questions.

Reviewer comments are repeated in *italic*, author replies in regular font.

Best regards,

Adrien Michel, on behalf of the authoring team.

*The manuscript provides a thorough investigation of modeled future temperatures in Swiss streams. Methods are well detailed and simulated streamflow for historic and future conditions are exhaustively detailed.*

*My main comments involve the length of the manuscript and the primary messages. The manuscript is almost too long, with certain side analyses partially detracting from more central messages of the manuscript.*

*On a related note, the abstract itself predominantly focuses on the future simulations (which – as admitted in the manuscript – have some potential limitations) while neglecting what I see as the more fundamental insights into hydrological process and the sufficiency of model structure. The manuscript had more nuance and deeper investigations into process than I was led to believe by initially reading the abstract.*

*Thus, I have two specific thoughts:*

1. *Possibly move Figure 7, 8, and 9 and some accompanying text to Supplemental section*

   We agree that the paper is rather long as it accommodates a substantial number of results. We will do our best during the revision process of the manuscript to shorten it. Figures 7, 8, and 9 could indeed be moved to the Supplementary along with part of the discussion only keeping the main message in the text. Reviewer 1 also suggested moving Section 5.3 to the Supplementary. We will consider these suggestions during the revision with the objective to shorten the paper and give the main messages better visibility.

   *Rewrite abstract to better emphasize insights into appropriateness of model structure and reduce emphasis on summary of future simulations. Similarly the conclusions section could also benefit from some shifting of prioritization of messages. In particular, there should be specific mention of that model does not allow for direct input of melt water into streams and that this led to overestimate of warming under historical conditions (but is believed to be less of an issue in the future as snow diminishes).*

   Thank you for this suggestion. Indeed, the paper is discussing quite extensively the models and their limitations. While the title gives some indication, this part is absent from the abstract as well as the conclusions. We definitely agree that these aspects

should be included. Here is how we intend to adapt the abstract, showing additions (in green) and deletions (in red):

"Rivers are ecosystems highly sensitive to climate change and projected future increase in air temperature is expected to increase the stress for these ecosystems. Rivers are also an important socio-economical factor impacting agriculture, tourism, electricity production, and drinking water supply and quality. In addition to changes in water availability, climate change will impact the temperature of rivers. This study presents a detailed analysis of river temperature and discharge evolution over the 21st century in Switzerland, a country covering a wide range of Alpine and lowland hydrological regimes. In total, 12 catchments are studied. They are situated both in the lowland Swiss Plateau and the Alpine regions and cover overall 10% of the country's area. This represents the so far largest study of climate change impacts on river temperature in Switzerland. The impact of climate change is assessed using a chain of physics-based models forced with the most recent climate change scenarios for Switzerland including low, mid, and high emissions pathways. The ability of such models for this application is discussed in detail and recommendations for future improvements are provided. Despite the identified limitations, the used model chain is shown to provide robust results. A clear warming of river water is modelled during the 21st century, more pronounced for the high emission scenarios and toward the end of the century.  At the end of the century (2080-2090), the median annual river temperature increase ranges between +0.9°C for low emission and +3.5°C for high emission scenarios for both Swiss Plateau and Alpine catchments. At the seasonal scale, the warming on the Swiss Plateau and in the Alpine regions exhibits different patterns. For the Swiss Plateau, the spring and fall warming is comparable to the warming in winter, while the summer warming is stronger but still moderate. In Alpine catchments, only a very limited warming is expected in winter.  The period of maximum discharge in Alpine catchments, currently occurring during mid-summer, will shift to earlier in the year by a few weeks (low emission) or almost two months (high emission) by the end of the century. . In addition, an important soil warming is expected due to glacier and snow cover decrease. These effects combined lead to a summertime river warming of +6.0∘C in Alpine catchments by the end of the century for high emission scenarios. Two metrics are used to show the adverse effects of river temperature increase both on natural and human systems. All results of this study along with the necessary source code are provided with this manuscript."

In a similar way, we propose to modify the first part of the conclusion as follows:

"This work presents the fist extensive study of climate change (CC) impact on rivers water temperature in Switzerland and, to the best of our knowledge, in Alpine areas. A chain of physics-based models has been used with 24 CC scenarios, spanning three different emissions pathways (RCP2.6, RCP4.5, and RCP8.5), and applied to two categories of catchments, namely lowland Swiss Plateau catchments and high elevation Alpine catchments. The work presented here required substantial

optimization work in the source codes of the models, which underlines the importance of good documentation, maintenance, accessibility, and collaboration around model source codes, which is often undervalued.

We demonstrate the ability of the developed model chain to reliably simulate the water temperature of a variety of catchments over Switzerland despite some limitations being identified. The results obtained for water temperature and discharge for the near future with 21 CC scenarios are coherent with past and current observations in Switzerland and in central Europe (Moatar and Gailhard, 2006; Webb and Nobilis, 2007; Arora et al., 2016), and with other modelling studies using the same forcing scenarios over Switzerland.

This work shows that the computation of the temperature of water flowing into the stream network is a critical factor in Alpine catchments and that omitting the cold advection from snow and ice melt (such as in the used HSPF approach) leads to an overestimation of the summer water temperature over the historical periods. Two other approaches were tested for the inflow temperature but led to worst results. Other aspects of the physical models used such as the impact of using a lumped approach versus a discretized approach for water routing, or a simple in-stream routing computation versus a more complex one, together with the calibration period length are tested. Therefore, this work offers thus a solid basis for future work on physics-based hydrological models.

Based on our modelling results, expected CC impacts on river water temperature in Switzerland can be summarized as follows: […]"

---

## Author Response (AR1)

Dear Editor, Dear Reviewers,

Please find below a detailed answer to the reviews along with the changes made to the manuscript. Answers to the editor and to both reviewers are included in the present document. Comments from the editor and reviewers are shown in light gray italic, our first answer (before editor decision) in black, and modifications performed after the decision of the editor in blue. In the blue parts, all figure, table, page and line numbers refer to the new version of the manuscript without track changes, unless stated otherwise. In the track-change version of the manuscript, comments have been added to indicate the new location of the text in case long parts of the text have been moved.

During the revision process, we addressed all comments, and the vast majority of them led to additions or clarifications in the text. Many additional references have been included. On the other hand, a substantial volume of text has been moved to the supplementary material to better focus the main text on the key methodological aspects and results. As a result, the text has been shortened by 10% and the number of Figures reduced from 12 to 8.

Despite this shortening, we were able to concentrate more on the work performed with the models, *i.e.* a clear emphasis on two aspects: 1) the validation of the models for such a study, and 2) the results obtained for climate change, as suggested by the second reviewer. Many parts of the text have been clarified following related remarks of the first reviewer and details have been added where necessary as suggested. Supplementary material has been significantly enhanced to support our work and answer to the concerns of reviewer 1.
All these modifications are documented in the track-change version of the manuscript.

We are aware that this paper is still voluminous as a lot of important methodology and results are presented at once. However, we think that all these elements are well connected forming a concise and complete study which we believe should not be chopped in several part. With the modifications performed following the very helpful remarks of the reviewers, we have now an improved manuscript much easier to read and more focused on the key points, while many minor aspects are now presented in the supplementary material for interested readers.

We hope that these substantial revisions are satisfactory for the reviewers and the editor.

Best regards,

Adrien Michel, on behalf of the author team.

**REPLY TO EDITOR**

*Dear authors,*

*Your manuscript has received two thoughtful sets of feedback from two reviewers. Both have noted the merits of your article and its contribution to the literature. Both have also identified some areas for improvement. In particular, Reviewer 1 noted several missing details in the study, and both reviewers have recommended shortening the manuscript. I agree with both reviewers that the manuscript could benefit from moving materials that are not central to the story to a Supplementary Materials section. I also agree with Reviewer 2 that refocusing the abstract and conclusions will benefit the impact of your study. I strongly encourage a thoughtful revision of the conclusions section.*

*Building from your thorough reply to these comments, I ask that you incorporate the changes recommended by the reviewers and submit a revised manuscript for consideration. I look forward to receiving your revision.*

Dear Editor,

Following the pertinent and helpful comments and recommendations of the two reviewers, our manuscript has been substantially revised. Many sections of the paper have been moved to the supplementary material. Additionally, several elements have been added in the paper in order to clarify the points raised by the reviewers. Both reviewers suggested to remove Section 5.3. After revision, we estimate this section to be relevant since it brings an important piece of information explaining the different model's sensitivities to increasing air temperature observed between lowland and Alpine catchments. However, this section has been improved and many related points mentioned before in Sections 5.1 and 5.2 have been integrated in this section to emphasis the relevance of the points discussed and ease the reading.

Despite many little additions, we have been able to shorten the paper and remove one third of the figures. The revised manuscript is much better focused on the main findings. Reviewer 1 raised some concerns about the HSPF groundwater temperature scheme used in the model. We delivered many arguments in our answers below to justify the robustness of our approach and implemented the most relevant part of it in the paper and supplementary material. At the same time, we tried to keep the additions in the main text short and concise.

Finally, following your recommendations, the abstract and conclusion sections have been carefully revised.

Best regards,

Adrien Michel, on behalf of the author team.

**REPLY TO REVIEWER 1**

Dear Reviewer,

We thank you very much for your constructive and instructive comments on our manuscript. We are very well aware that this review has been a significant time investment and therefore especially appreciate the reviewer's feedback and commitment. Please find below a detailed answer to all the points of this thorough review. We hope that these answers together with the corresponding proposed modifications of the manuscript will provide the requested clarifications and consider all of the reviewer's useful suggestions.

Best regards,

Adrien Michel, on behalf of the author team.

Reviewer comments are repeated in light-gray *italic*, author replies in black regular font.

*The manuscript by Michel et al details a study that uses a chain of models to simulate river temperatures in 12 Swiss catchments that encompass two main landscape types: alpine and lowland. They used downscaled climate change projections in a suite of models that simulate glacier, snow, soil temperature, runoff, and river temperature dynamics. They assess the ability of the modelling chain to simulate historic conditions using observations from around 2005-2018. They conclude that river temperatures will increase for both catchment types under climate change; however, the magnitude and seasonality of increases will differ. The lowland (Plateau) catchments are expected to see relatively uniform river temperature increases throughout the year, with slightly higher increases in summer. In contrast, alpine catchments are expected to see small increases in winter river temperature, but large increases during summer due to an earlier shift in peak discharge associated with lower amounts of snow and glacier melt contributions.*

*Overall, the manuscript is relatively well written (although a number of typos and grammatical errors need to be corrected), and the logical development and presentation of the study generally makes sense. This was clearly a significant logistical effort to build and work with this chain of models. I can also appreciate the challenge in synthesizing these efforts. Because of the amount of work presented, and that much of it builds on previous efforts (i.e., the specific model components), I found it an overwhelming piece of work to digest. I appreciate the amount of supplementary material included, but the decision to not always reference that material directly in the text (supposedly to 'alleviate the text'?) makes it very difficult for the reader to navigate the immense number of results shown. Despite the considerable amounts of results included, the key findings from these modelling exercises don't strike me as overly complicated; therefore, I think there is a great opportunity to streamline the manuscript by reducing the amount of results presented while still providing support for the key findings in this study.*

We are aware that this manuscript is rather long and extensive. It has been discussed at some point to maybe split it into two papers (one on the dataset and one on the analysis), but we preferred to keep it as one comprehensive study rather than chopping it down into fragments.

We agree that there is potential for shortening and we will try our best during the review process to make the story more concise. Some suggestions for shortening are presented below and in the answer to Reviewer 2. At the same time, the reviewer also suggested some additions and clarifications. We will do our best to include this information without further extending the paper. Regarding more frequent references to the supplementary material, we will add these to the text.

As mentioned in our initial reply above, we have significantly reduced the paper length by moving many parts of secondary importance to the supplementary material. Especially, all details about the tests of the different modules, along with the detailed discussion about the climate change scenarios validation (Sections 4.3 and 4.4 in the first version of the manuscript), have been moved to the supplementary material. Former Section 3.4 about the models' upgrades has been removed and included condensed as two sentences in Section 3.1, along with a reference to a recently published PhD thesis containing a detailed description of all the models' upgrades. The manuscript now emphasis only the ability of the models used to succeed in such an application, but leaves many details in supplementary material for interested readers. Following your recommendation, we now refer more consistently to supplementary material elements in the main text.

In parallel to these shortenings, we could enhance and clarify many parts of the paper addressing the reviewer's concerns. All details of these changes are given in the following replies below. The manuscript text was shortened by 10 % (since also many additions were requested), and is now more focused on the key points, clearer, and hopefully easier to read.

*1. Describing the model chain and key asumptions/limitations*

*Keeping track of the various model components, what they do, and how they were individually and collectively calibrated is challenging. Perhaps a flow chart or a diagram detailing how the models interact, what components are tested against observations and/or what parameters are calibrated would help readers? For example, I was really confused about the time periods used to calibrate the individual models. Alpine3D was calibrated to 2012-2018? StreamFlow was calibrated to 2012-2014, but then validated to 2015-2018? So this is not an entirely independent validation since Alpine3D was already calibrated to that period? In addition, the 'Validation over climate change period (Section 4.2)' was validated for 2005-2015 - which encompasses the period used for calibrating the model. So again, this is not an independent validation? I'm likely misunderstanding this workflow, which is why a better description of the steps taken might help the reader.*

Thank you for this suggestion. Please find below a first draft of a figure summarizing the models' interactions and the various steps (Figure 1). To answer your question, there is no general calibration of Alpine3D. We perform a few model runs of Alpine3D over the whole period to adjust the vertical precipitation lapse rate in order to achieve a correct mass balance over the year. We agree that this choice is not validated afterwards. A similar procedure was used in Gallice et al. (2016) and Brauchli et al. (2017).

Figure 1 below has been added into the manuscript as Figure 2. Regarding Alpine3D, Section S5 has been updated in order to clarify how the precipitation lapse rate is chosen.

[Figure]

Figure 1 – Details of the models' workflow. The calibration and validation periods indicated hold for all catchments except for the Eulach catchment (where the periods 2015-2016 and 2017-2018 are used). Figure adapted from Gallice et al. (2016)

Regarding Section 4.2 "Validation over climate change period", the objective here is to compare the model output for runs forced with climate change scenarios compared to runs forced with measurements, and consequently we do not consider it an issue if a few years overlap with the calibration period. Maybe the section could be renamed "Evaluation of climate change scenarios", because this is the real purpose of this section. Indeed, climate change scenarios have been downscaled to perform this work. Both the "base" climate change scenarios CH2018 and the downscaled version come with a few shortcomings and warnings (detailed in Section 2.4 and in Michel et al., 2021), and the purpose of this section is to ensure the correct usage of the models when forced with these time series (which are not the time series used for calibration). In order to shorten the manuscript, we propose to move this section into the Supplementary.

This section has been renamed "Validation of climate change scenarios" and moved to supplementary material as Section S8. The part describing the use of 10-years time periods has also been moved to the supplementary material (Section S7). They are now referenced in Section 2.4 (p.7 l.29-34 and p.7 l.21-22, respectively) in the main text.

*In addition, key details or assumptions made by the various model components are not really addressed - although these assumptions can be critical for interpreting the model results. Instead, the reader is often referred to other studies and publications to get these important details. I recognize that the authors don't want to duplicate information already contained in other papers; however, I think the key aspects and assumptions of those models relevant to this study need to be presented in this manuscript.*

Indeed, we tried not to extensively repeat the existing literature for the sake of length of the manuscript. Alpine3D is entirely physics-based and thus no particular assumptions are made, however, there are still some relevant issues (e.g., no snow transport by wind or no partially snow-covered pixels, we will mention this in Section 3.1).

Regarding StreamFlow, the main assumption concerns the soil water temperature and we think this is already largely discussed in Sections 3.3, 4.1.2, and 4.4. The two-linear reservoir approach used is also an approximation; submitting to a hydrology journal, we omitted the details on this well-known concept but we will refer explicitly to a key reference such as Perrin et al. (2003). The incoming shortwave radiation would deserve more attention. We discuss it further below and make a suggestion on how to integrate it in the paper.

Section 3.1 has been updated accordingly, see p.10 l.27-29. Section 3.2 has been significantly updated (p.11, l.24 to p.12 l.4), see below for the part about solar radiation. In addition. Section 5.3 about the limitations of the model has also been significantly improved, see in particular p.29 l.23-34.

*Also, were any 'spin-up' or 'warm-up' periods used for any of the models? If not, why?*

Yes, there is a spin-up period. Thank you for having noticed this missing information. Alpine3D is always started for the month of July, with a temperature profile approximated from previous simulations, and model results are used in StreamFlow only from October on. There is thus a

3 months period for the soil temperature profile and water content to equilibrate. StreamFlow always uses a period of two years as a spin-up. This is achieved by running the model over the first two years of the time period, and then re-starting from the beginning of the time period from either the real calibration or simulation. We will clarify this point by adding the details above in Section 4.1.1 where we present the time periods used.

Section 4.1 has been updated to reflect these points (p.12, l.14-15 and l.22-25)

*2. Clarifying the advection term associated with runoff*

*Probably a consequence of the information overload outlined in my point #1 above, I'm struggling to understand if and how the advection term associated with hillslope/land runoff is treated in these models. Alpine3D simulates spatially distributed soil temperatures and water available for runoff. StreamFlow sums the water available for runoff for all the Alpine3D cells draining to a stream reach of interest?*

Yes, each stream reach is associated with a sub-watershed. Figure 1 above, that will be added to the manuscript, clarifies this.

Figure 1 has been added as Figure 2 in the revised manuscript (see reference p.9 l.1), and we now explicitly mention the heat advection in the text (p.11, l.14).

*This water available for runoff is assigned a temperature by one of three methods: 1) the energy balance approach in Comola et al (2015), 2) the HSPF algorithm, or 3) a soil temperature value (I am assume taken from Alpine3D?). Very little detail is provided in the text or in the supplementary material about the details and differences in these approaches (other than different RMSE reported in Table S7) and the authors conclude that the HSPF is the most consistent, so they use that for all subsequent model runs. However, I'm confused by the statement that 'in the HSPF scheme, the soil temperature has a less important impact than in the other schemes (soil temperature is only needed for heat conduction between water and river bed).' In this statement, does this mean that the soil temperature output from Alpine3D is only used for the bed conduction term, but in the other schemes it is used for something else? I get that the soil temperature from Alpine3D is used to set the runoff temperature in scheme 3, but how is it used in scheme 1 (the Comola approach)? Also, it sounds like Alpine3D simulates soil temperatures at different depths - so which depth is used for scheme 3?*

Yes, in the approach 3), the soil temperature at a given depth (obtained from Alpine3D) is used in StreamFlow to set the temperature of the water entering the river. In the approach 1), the temperatures of the two reservoirs are obtained as follows (see full details in Comola et al., 2015 and a summary in Appendix A2 in Gallice, 2016):

$$dT_U/dt = I_U/S_U * (T_S-T_U) + (T_S-T_U)/k$$

$$dT_L/dt = I_L/S_L * (T_S-T_L) + (<T_S>-T_L)/k$$

Where $k$ is a calibration parameter corresponding to the characteristic time of thermal diffusion, $T_s$ the soil temperature at a chosen depth (obtained from Alpine3D), $T_U$ and $T_I$ the water temperature in the upper and lower reservoirs, respectively, $I_U$ and $I_L$ the infiltrating water to the corresponding reservoir, $S_U$ and $S_I$ the current reservoir storage, and <...> denotes the average of the value over the simulation period. In both equations, the first term corresponds to the energy flux from the inflow water to the reservoirs, and the term on the left-hand side to the diffusive heat exchange between the water and the soil. We will add this mathematical description in the supplementary material. By model design, the soil depth used to take the soil temperature in the approaches 1) and 3) is the same as the depth used to compute the heat exchange with the river bed, which can be interpreted as a limitation. However, since we went to the HSPF approach where the soil temperature is used only for heat exchange with the riverbed, this problem disappears.

For all 3 approaches, the model is calibrated using soil depths between 0.5 and 3 meters, at intervals of 0.5 m. The soil depth leading to the best results is retained. The temperature calibration being computationally intensive, an iterative approach is used, i.e., only some depths are tested and additional depths are added if necessary, explaining why different depths are shown in the figures below. Figure 2 shows the details over the calibration period for two non-alpine catchments, and Figure 3 for two alpine catchments. From these figures we see 1) that the difference resulting from the choice of the soil depths is almost negligible for the HSPF approach, and 2) that the HSPF approach is performing better. The particular situation of the alpine catchments is discussed below when answering to your remark concerning the choice of HSPF instead of the approach 1) of Comola et al. (2015). An important note for the interpretation of the plots below is that all results come from a separate calibration. So, if most of the difference seen can be attributed to the chosen soil depth, part of the difference can also be due to the calibration, and this ratio is hard to quantify. The discussion above along with Figures 2 and 3 below will be added in Supplementary material Section S5. The main message will also be better explained in Section 4.1.1.

Supplementary material Section S5 has been significantly revised to clarify the procedure. Also, soil temperature is now explicitly mentioned when describing the Comola et al. (2015) approach in Section 3.3 (p.11, l.16-17).

*Also, is it appropriate to use the Alpine3D soil temperature for the channel bed temperature? The channel bed has entirely different upper boundary conditions than the terrestrial parts of the catchment and it seems inappropriate to use the Alpine3D soil temperatures to represent channel bed temperatures (especially since the authors note that some of these catchments experience substantial flow losses along their network; therefore, the stream bed temperature will likely be influenced by river water infiltrating the subsurface).*

Indeed, this is a rather rough approximation. However, as shown in Figures 2 and 3, for the HSPF approach, the effect of the chosen depth is almost negligible, despite of the soil temperature cycle having a very different annual amplitude at 0.5 meter or at 2.5 meters depth, suggesting that the actual effect of the riverbed heat exchange is in general not a major energy flux. In addition, there is a calibration parameter (the amount of energy transferred between the river and the riverbed by degree kelvin of temperature difference) that plays an important

role and this calibration parameter might also be able to correct for the fact the we simplify the system by taking the soil temperature for a terrestrial part and ignoring the upper boundary condition imposed by the river. The values of this calibration parameter are indicated in Figures 2 and 3, and the wide range of values obtained along with the trend observed with soil depth support this hypothesis of the calibration parameter correcting for the simplified soil temperature used. As for the point above, in the revised version this will be detailed in Section S5. In addition, in Section 3.3, where StreamFlow is presented, will discuss this aspect.

Supplementary material Section S5 has been significantly enhanced and now covers this aspect, including Figures 2 and 3 below. Section 3.3 now also mentions that the same soil depth for taking the temperature is used both for the streambed heat exchange and the soil water temperature schemes using soil temperature (p.11, l.30-31).

[Figure]

Figure 2 – Details of the water temperature calibration phase with different soil depths used for the soil temperature. Calibration for the Broye catchment (top) and the Kleine Emme catchment (bottom), both located on the lowlands. The three approaches available to compute the water temperature in the soil are used: 1) the energy balance formulation (EB, Comola et al., 2015), 2) the HSPF formulation (Bicknell et al., 1997), and 3) the soil temperature approach (ST, Gallice et al., 2016). The root mean square error (RMSE) and the calibrated value obtained for the river-streambed heat transfer coefficient are indicated in the legend.

[Figure]

Figure 3 – Details of the water temperature calibration phase with different soil depths used for the soil temperature. Calibration for the Landwasser catchment (top) and the Kander catchment (bottom), both located on the Swiss Alps. The three approaches available to compute the water temperature in the soil are used: 1) the energy balance formulation (EB, Comola et al., 2015), 2) the HSPF formulation (Bicknell et al., 1997), and 3) the soil temperature approach (ST, Gallice et al., 2016). The root mean square error (RMSE) and the calibrated value obtained for the river-streambed heat transfer coefficient are indicated in the legend.

*The decision to use the HSPF approach seems like it could have important implications for the climate change modelling, particularly for the alpine catchments that see a decrease in snow cover. As the work by Yan et al (2021) highlights (and some of the calibration issues in this study also suggest), this runoff advective flux can be important in snow-dominated catchments. Based on a quick search, I see that the HSPF algorithm doesn't account for the presence of snow and therefore may be unsuitable for looking at climate change impacts in snow influenced catchments (see Leach and Moore, 2015). Assuming this is the same HSPF algorithm (I suspect it is), could this be partly why the modelling is under-estimating spring/summer temperatures for the alpine sites? Why not use the Comola or Alpine3D approach which, I presume (but no details on these schemes are provided) can account for the influence of snow cover on runoff temperatures? If it can account for the snow influence, I would argue that would be a preferable approach even if the calibration metrics aren't as good as the HSPF values, since it should better extrapolate to future conditions.*

The question of snow cover is indeed relevant. As presented in Section 4.4, we think this is the main issue leading to the overestimation of the water temperature in summer. As you correctly mention, the HSPF approach does not take into account snow cover, and Leach and Moore (2015) state that: *"The HSPF model, original CEQUEAU function, and LARSIM-WT were the most successful approaches at estimating throughflow temperatures at Griffith Creek with prediction rates of 53%, 33%, and 31%, respectively. However, a major drawback of these approaches is that they do not account for snow, which limits their use as a predictive tool to under- stand stream temperature response to changes in climate and land cover, and corresponding impacts on snowpack dynamics.".* However, among the 6 different approaches tested for a snow-covered area, they also show that: *"Of the models considered here, the HSPF interflow equation was the most successful at predicting throughflow temperatures. HSPF predictions fell within the range of observed throughflow temperatures 53% of the time and only overestimated throughflow temperatures by a maximum of 1.38 °C.".*

As discussed in the manuscript in Section 4.1.1, the only influence of snow on water temperature when using the HSPF approach, in addition to the mass added, is the fact that snow cover influences the mean sub-catchment soil temperature, which is then used to compute heat transfer (advection) to the river. During the melt season (when advective flows are important) it leads to good results since air and soil temperature are still low, as shown in Figure 3 above, but not in summer.

In the energy balance approach of Comola et al. (2015) detailed above, the soil temperature plays a more important role. Since the soil temperature in Alpine3D is heavily influenced by the presence of snow, we can expect it to lead to better results. However, in this approach, there is no direct accounting for the water originating from snow melt or from precipitation (this would not be straightforward to implement in the current versions of the models). In the lowlands catchments (Figure 2), performances are similar between the HSPF and the energy balance approaches, except that the energy balance approach exhibits a too marked temporal variability compared to measurements. In alpine catchments (Figure 3), this approach shows clear problems in correctly reproducing the water temperature. In the Landwasser catchment, the water temperature in winter is clearly overestimated and the variability during the melt season is largely overestimated. In the largely glacierized Kander catchment, the summer

stream temperature is underestimated. This is probably explained by the usage of the mean soil temperature over the whole sub-catchment to compute the energy balance, i.e., all the water originating from the glaciated area enters the rivers mostly at 0°C, while in reality the water already warms while flowing around the rocks. Overall, if the HSPF approach underestimates the snow cover effect, the approach of Comola et al. (2015) seems to overestimate it. This is also observed by Gallice et al. (2016). Figure 3 also shows that the EB approach or the simple approach of taking the soil temperature as proxy for soil water temperature gives very similar results.

Despite the quality metric is only slightly lower than obtained with HSPF, we see in Figure 3 that the EB approach of Comola et al. (2015) presents major issues, which we estimate to be more important than the problem of summer overestimation of the HSPF approach, explaining why the HSPF approach has been used. In the current version of the manuscript only the RMSE is discussed. In order to fully clarify the choice of the HSPF approach, we will add the present Figure 3 into the supplementary material.

As shown in the two studies you suggest (Leach and Moore, 2015; Yan et al., 2021), this question of subsurface water temperature is not a straightforward task and the HSPF approach used has some obvious weaknesses. However, we believe that we sufficiently emphasize this in the manuscript and we are very clear about the associated limitation. In addition, the snow cover effect accounted in the model through streambed conduction has an impact, as shown by the increased warming in Alpine catchments expected with climate change compared to the lowland ones. Nevertheless, we agree that both the abstract and conclusion do not emphasize enough on these modelling aspects, and we will revise these accordingly (see comments from Reviewer 2 and our answer).

Thank you also for pointing us to the recent work of Yan et al. (2021), which we were not aware of at the time of submission. Indeed, that paper shows that snowmelt has an impact on water temperature. However, in our interpretation of their results, Yan and colleagues were primarily considering the upper and lower bound of the contribution of snow melt on water temperature. For infiltrating water temperature, they use either 0°C for water originating from snowmelt (which is clearly a lower bound and neglects any warming before entering the river), or mean annual air temperature otherwise (which is clearly an upper bound during the melt season). Indeed, when they turn on or off the value of 0°C for snowmelt the difference is significant, as one could expect using such bounds for infiltrating water, but we note that the actual effect must be in between those two scenarios and therefore likely to be smaller.

We also think that the analysis in the work of Yan et al. (2021), does not confirm that the approach would lead to more accurate temperatures in the melt season, since the model is calibrated using summer temperature only, and the water temperature is not validated for the calibration station. In addition, the model is calibrated for stations situated rather low in the catchment and when we go to the stations upstream (see Figures 3 and S3 in the paper), we clearly see that the simulated water temperature is underestimated in winter and fall and overestimated in spring (similar to the issues we have for the Landwasser when using Comola's approach shown in Figure 3 here). The metrics used for calibration are not directly comparable (they use MAE, we use RMSE, and MAE <= RMSE). The MAE obtained by Yan

et al. (2021) for calibration centered in one season, leads to MAE which are similar or larger than the RMSE we obtain (and RMSE is highly penalized by the summer overestimation in our results). Based on this, we conclude that the model used in Yan et al. (2021) is not expected to show a big advantage in the catchments in our study.

Their main results regarding the impact of snowmelt are shown in Figure 3 in their paper. As we argued before, the values obtained in Yan et al. (2021) represent upper and lower bounds and we also would argue that the effect is overestimated, since the model does not seem to have been recalibrated for the case without the temperature effect from snowmelt. Since we do calibrate our model chain with only a weak effect of snowmelt on water temperature, the calibration may partially compensate for that.

We agree that the snowmelt temperature is an important parameter and its neglection is a drawback leading to a summer water temperature overestimation that we amply discuss in the paper. We will include the study by Yan et al. (2021) in our manuscript and briefly summarize the elaborate explanation given above. However, we think that the study by Yan et al. (2021) is not directly applicable to our study approach, as detailed above.

In Section 3.3, it is explicitly mentioned that the Comola approach "uses the energy balance between groundwater and soil temperature" (p.11 l.16-17), which is an important clarification for the reader. Sections 4.1.3, the part about Alpine catchments, and Section 4.4 (both old version numbering) have been merged into Section 4.2.2 in the revised manuscript (p.16 l.1 to p.18 l.14). The content of this section has been updated in order to reflect the discussion above, and at the same time it has been shortened and further clarified. The suggested references have been added (p.17 l.6 and l.11), along with other new references. This is complemented by the extended discussion of Section S9 and by the new figures and discussion added in Section S5 (where water temperature schemes are discussed and where we show in detail why we use the HSPF approach). Finally, abstract and conclusions have been completely reworked to put more emphasis on the modelling aspect. A new paragraph has also been added for this purpose at the end of Section 4.2.2 (p18. l.10-14)

*3. Other modelling studies from mountainous snow-dominated environments*

*The authors primarily reference other Swiss studies throughout the manuscript. There have been other studies looking at hydrology and river temperature response to potential climate change scenarios conducted in mountainous snow-dominated environments. I'm familiar with some of the work from western North America. Some examples include: Null et al 2013, Leach and Moore 2019, Yan et al 2021. In particular, Yan et al (2021) seems highly relevant here. I think it would enrich this manuscript to incorporate the findings from some of these studies in the introduction and discussion (there are some interesting similarities and differences between the findings from those studies and the results presented here).*

Thank you for these suggestions. By the time of our manuscript submission, we were not aware of the very recent publication of Yan et al. (2021), discussed above. The work of Leach and Moor (2019) was in fact one of our motivations to use physics-based models for our study and unintentionally got forgotten to me mentioned. This will be of course corrected. We will

include more comparisons with results outside Switzerland (e.g. Morrison, 2002; Null et al., 2013; van Vliet et al., 2013; Du et al., 2019; Leach and Moore 2019).

Many references have been added in Section 5.1 (p.19 l.7 and p.20 l.8-10), 5.2 (p.27 l.12-18) and 5.4 (p.28 l.12-16 and p.29 l.15-16), corroborating our findings.

*4. Key assumptions on river temperature modelling*

*Maybe these details are contained in the StreamFlow references, but I was surprised by the lack of discussion on potentially key assumptions around some of the river temperature modelling. In particular, there is almost no details or discussion about the role of riparian vegetation and its influence on radiation exchange and the sensible and latent heat fluxes. The manuscript mentions that topographic shading is taken into account (at least for Alpine3D, it's not clear if this is also the case for StreamFlow) - is that the only source of shading for these rivers? Maybe that is the case? If so, I would recommend clarifying this point. If not, it seems prudent to discuss the potential issues that ignoring the role of riparian vegetation might have on the modelling. Along these lines, I also wonder if a discussion on potential land cover changes in these catchments over the next decades, and how they might also influence river temperatures, might be worth including? This is touched on a bit, but could be expanded.*

Thank you for bringing up this point. Indeed, riparian vegetation is a topic absolutely worth investigating. Let us first answer the questions related to how the model takes riparian vegetation into account. We will add these clarifications into the sections where the models are described (3.2 and 3.3).

In Alpine3D, the topographic shading is taken into account to compute the radiation. In addition, a two-layer canopy module is used to compute, among other, the vegetation shading. The forcing grids used for the hydrological simulation in StreamFlow take both of these aspects into account (note that the canopy does also modify the wind speed). However, this has some limitations. Both topographic and vegetation shading are computed from elevation or land use grids at 500 m spatial resolution. The local small-scale topography of the river is ignored (and in alpine areas, rivers can be some meters below the surrounding terrain and experience some shading). Regarding vegetation, the local riparian vegetation might also be ignored since pixels at 500m resolution from the CLC dataset are used. In mountainous area, this might also lead to underestimated shading, while in the lowlands the shading might be overestimated when a large river crosses a forested pixel (and will thus be considered as totally shaded since the canopy module in Alpine3D does not compute any shading projection).

We initially thought of a possible overestimation of radiation to explain the overestimation of water temperature we observe in summer in alpine catchments. However, the detailed analysis presented in Section 4.4, in particular, the energy fluxes show that the warming events are quite sudden and not related to the radiative fluxes, explaining why we do not consider it as the first candidate for the error of the model. An extended discussion on this matter is presented below when answering another reviewer question.

Given the current extent of our manuscript, a longer discussion on riparian vegetation and potential impacts of land use change was considered beyond the scope of this paper. However, we will add a paragraph about this in Section 5.4 in which we address this limitation of the model. The limitations of the topographic and vegetation shading using 500m grid cells will be mentioned, together with the fact that this study assumes that the land cover will not change throughout the 21st century. This will also be mentioned in Section 4.4 when discussing the water temperature overestimation in summer (again see discussion below).

Overall, both the impact of snow cover and of the vegetation are primordial questions to be addressed in future studies and we hope that the models' setup established for this work will serve as starting point for related discussions.

Section 3.3 has been updated to clarify the treatment of radiation in StreamFlow (p.11 l.33 to p12 l.4). Riparian vegetation is also mentioned in the new Section 4.3, where the summer temperature overestimation is discussed (p.16 l.27-31), in Section 5.4 (p.29 l.23-26), and extensively detailed in Section S9 in the discussion about radiation.

*Specific comments:*

*P1L3: Perhaps expand or give an example why rivers are important socio-economical factor.*

We can add the following in the abstract (paraphrasing P1L20-22): "The literature clearly identified several sectors which are vulnerable: agriculture, tourism, electricity production, and drinking water supply and quality (e.g., Hock et al., 2005; Barnett et al., 2005; Schaefli et al., 2007; Bourqui et al., 2011; Viviroli et al., 2011; Beniston, 2012; Hannah and Garner, 2015)."

The abstract has been revised accordingly.

*P3L5: I would replace 'attributed to the' with 'associated with an', since it is fairly well established that although air temperature is often correlated with water temperature, air temperature itself, via the sensible heat flux, is not often a key control.*

We will change this.

Done, see p.3 l.1

*P6 Section 2.3: How were data from various met stations used as inputs to the models? Lapse rates? Thiessen polygons? Some other adjustment? Ok - I see this is provided in Section 3.2.*

Yes, we can add a cross-reference to Section 3.2 in Section 2.3.

Done, see p.5 p.30-31

*P9 Section 3.3: How are energy exchanges at the stream-atmosphere interface dealt with? Is radiation exchange adjusted for riparian conditions? Are the land-based meteorological measurements adjusted for above-stream conditions for the sensible and latent heat flux calculations? The reader is directed to Gallice et al 2016, but some general overview on this aspect should be included here.*

The point about vegetation is already discussed above. For both latent and sensible heat fluxes, empirical equations are used (see Hannah et al., 2004; Haag and Luce, 2008; Magnusson et al., 2012, and Comola et al., 2015). These equations use the air temperature without local correction. Although these equations are common and widely used, we will add a short paragraph giving a general overview and the relevant references.

See update on p.11 l.27-29. We added the main references. Since this approach was used in two published papers (Comola et al., 2015, and Gallice et al., 2016) and an extended description is given in Gallice et al. (2016), we prefer not to repeat these details here.

*P18 Section 4.4: The model's inability to reasonably simulate the extremely warm 2003 period seems to be a critical issue, particularly since this model is being used to simulate climate warming scenarios (the model seems to be clearly missing an important heat sink). The authors do a reasonable job of discussing this modelling error, but the justification for continuing with the climate change predictions is a bit confusing to me. It seems like the checks*

*(by comparing the 2014 and 2015 summer periods) doesn't really get at the heart of the matter in that it seems to be checking whether the model gets the right answer, but doesn't care if it is for the right reason or not.*

First, we want to clarify an important point: the problems with the year 2003 is only for the Alpine catchments. Indeed, Figure S3 shows that for the lowland Broye catchment the extremely warm year 2003 is very well simulated. We agree that the discussion should be clarified and we put additional emphasize on this in the revised version. Here, we provide a summary of the main points. First, by looking at all energy fluxes, we investigate the origin of this error. We show that the warming occurs in small upstream reaches and that the missing heat sink there is probably the absence of direct cold advection due to snow and glacier melt. In the future, the snow cover in summer is expected to greatly decrease or disappear, and the glacier melt will be reduced compared to nowadays, so this issue might be of lesser importance.

To ensure that the issue is not caused by another factor and that the overestimation will not grow in the future, we use the comparatively cold summer of 2014 (for which the model performs well) and the warm summer of 2015 (when the model overestimates the water temperature). Since the method used to downscale the climate change scenario keeps the interannual variability of past time series, the summers e.g., 2084 and 2085 will exhibit the same relative difference (but on a warmer baseline). We can then again compare the relatively cold and warm years to see whether the difference grows (suggesting that the overestimation would grow). The results of the comparison show that this is not the case.

In summary, we show strong evidence that the cause is the missing snow melt water (see also discussion below) and that the error does not increase in simulations of future scenarios. Since snowmelt will become less important in the future, we decide to go ahead with climate change simulations. For the sake of transparency, we provide an in-depth analysis (not present in some other published studies with similar issues), present arguments to support our statements, and in the conclusion mention that confirmation of these results by further studies would be beneficial.

The question about the summer 2003 has been clarified in Section 4.2.2 (p.16 l.16-21). Further, the discussion about the error has been clarified (Section 4.2.2, p.17 l.25 to p.18 l.4) and extended in Section S9.

*P32 Section 5.3: I think this section and analysis can be removed. The physics-based modelling exercise already highlights the differences in discharge and stream temperature response to climate change for the alpine and plateau catchments. I'm not sure what the statistical analysis adds and the hypotheses being tested with these analyses are likely not what is intended (see Greenland et al 2016 for a discussion on this topic). In particular, the conclusion made on P32L16 that 'changes in discharge have no impact on water temperature change' is clearly wrong when considered from first principles (except for very unique cases that would not occur in reality).*

We will shorten this section and move most of the content into the supplementary material. However, we still think that the statement on P32L16 is relevant. Discharge is a factor influencing water temperature, but it is not clear to which extent. In many models, the larger influencing factor is the air temperature, which is clearly the best explanatory variable for water temperature, while the discharge plays a more minor role (see e.g., Feigl et al., 2021).

Here, we do not discuss the direct link between discharge and water temperature, but the link between discharge trends and water temperature trends in a warming climate. In Michel et al. (2020) we show from historical data across Switzerland that observed changes in discharge are not related to observed changes in water temperature on the long term. On the short term, heat waves in central Europe are usually linked to very dry periods (Fischer et al., 2007), and assessing the interplay between low discharge and high-water temperature is thus like a chicken and egg problem.

What we show in this paper is that discharge trends have a weak to inexistant ability to explain future change in water temperature in the lowlands, while it is a more important factor in Alpine catchment. This is what we state in P32L16: "changes in discharge have no impact on water temperature change" when talking about the lowland catchments. Correlations and linear regressions do not mean causality, but to have causality we expect to see some correlation, especially for discharge where we would expect the impact to be linear (i.e., the link between mass, energy exchange and temperature would be linear). We find this result very interesting and worthwhile investigating. Indeed, not all energy fluxes will be influenced by discharge and these results raise interesting questions regarding the dominating energy fluxes governing water temperature. Again, this result is also supported by historical measurements (Michel et al. 2020). Section 5.3 is specifically devoted to this question of the link between the trends in the variable, rather to just the difference in the response to climate change between the lowland and Alpine catchments.

In summary, the complexity behind stream temperature leads to a situation where less water would not necessarily mean warmer water (at least not as the dominant factor), in contrast to what we could expect from simple thermodynamic laws. In order to clarify and remove any ambiguity, we suggest to reformulate P32L16 to: "This confirms the hypothesis that changes in discharge is weakly correlated with water temperature change in the lowlands regions".

The question of the interplay between changes in discharge and in water temperature could be the core of a separate publication in the future with a focus only on this question, using historical measurements, i.e., an in-depth analysis of what we started in Michel et al., 2020, and the simulations results produced here.

After much reflection, we prefer to keep Section 5.3 because we think it is a relevant piece of information to explain the difference between the temperature rise expected in the Swiss Plateau and in the Alpine catchments. However, we removed the correlation analysis spread between Sections 5.1 and 5.2 and Table 6 since it was partly redundant with Section 5.3. Section 5.3 has been revised significantly to better integrate in the paper and show it's added value.

*P19L8-9: Missing relatively cold runoff inputs seems like a plausible reason for the model overpredictions (see my general point #2 above); however, would we expect the mechanism proposed in the previous paragraph (snow and glacier melt flowing over frozen or saturated soils) to be occurring during summer periods? Wouldn't it be more likely that HSPF is simply simulating warmer runoff temperatures than is actually occurring? Or maybe cold groundwater inputs (perhaps from a more regional source) are not being accounted for in the models? Or could not accounting for riparian vegetation shading be a factor here?*

This question has partially been discussed above. The starting point of our reasoning was the fact that the summer temperature overestimation only occurs in Alpine catchments while the model gives very good results for the lowland catchments. This allowed us to already exclude some candidates (e.g., programing errors in the model source code). The groundwater hypothesis is discussed in Section 5.4 as one of the shortcomings of the model (by groundwater we mean here water below the water table, i.e., the saturated zone, not subsurface runoff). The fact that the problem with summer temperature overestimation occurs only in Alpine catchments, regardless of a known or unknown interaction with groundwater (e.g., the Landwasser, see Epting et al., 2021) and not on the lowland areas, again regardless of any groundwater interaction, shows that the groundwater interaction is not a plausible explanation to the problem of stream temperature overestimation during summer.

Riparian vegetation has already been discussed above. For alpine rivers, which are usually smaller than those of the lowlands, not considering the local vegetation may have a stronger impact on stream temperature than on the lowlands. However, for example for the very small catchment of the Rietholzbach on the lowlands, explicit consideration of riparian vegetation would also be important. Figures 4 and 5 show orthoimages of the Rietholzbach catchment and of the Alpine Lonza catchment, and Figure 6 shows the plot of the water temperature during the calibration and validation phases of the model.

From the orthoimages, we see that riparian vegetation is more present in the Rietholzbach catchment. However, Figure 6 shows that in this catchment the model does not produce the sudden water temperature overestimation simulated in the Lonza catchment. These results suggest that the lack of riparian vegetation shading is not responsible for the overestimation we have in alpine catchments. However, in the Rietholzbach catchment there is indeed a slight overestimation of the water temperature during summer and fall (but all over the period and not as peaky as in the Alpine catchments) which could be attributed to the lack of vegetation shading. As mentioned above, riparian vegetation will be discussed in Section 5.4 as one of the limitations of the model, and also in Section 4.4 as a factor probably contributing to the summer overestimation in Alpine region, but not as the main driver of the sudden peaks observed.

Topography and its treatment have also been already discussed. For completeness, we show in Figure 7 the same orthoimage for the Lonza catchment as in Figure 5 but adding also the DEM used in Alpine3D to compute the shading. Despite the coarse resolution of the DEM, we see from the figure that the topography is still reasonably well represented to allow for a correct shading on the main reaches of the river (for the large-scale topography).

In order to push further the question of the radiation impact (and the potential errors arising from topographic and riparian vegetation shading), we made a new run of the model for the Landwasser catchment with the artificial situation of zero incoming shortwave radiation. The results are shown in Figure 8. Note that StreamFlow has not been re-calibrated for this run, but that it uses the same calibration parameters as the other runs performed for the Landwasser catchment. The goal here is to only assess the model sensitivity to radiation and to see whether the high-water temperature peaks observed in summer disappear when incoming solar radiation is removed.

[Figure]

Figure 4 – Rietholzbach catchment (red) and river course (blue). Source: Swisstopo

[Figure]

Figure 5 – Lonza catchment (red) and river course (blue). Source: Swisstopo

[Figure]

Figure 6 – Top two panels: Measured (black) and simulated (red) water temperature for the Alpine Lonza catchment (first plot) and difference between the measured and simulated water temperature (second plot). Bottom two panels: Same as top, but for the lowland Rietholzbach catchment.

As expected, there is a general bias toward colder temperature when removing solar radiation. A first look at Figure 8 could suggest that the summer overshoot might be corrected by removing radiation, but when considering the summer 2014 we see that we have now a negative bias in relatively cold summers. The second panel of Figure 8 is more informative. Here we see that most of the time the model error in the summer (black line) is not at all correlated with the temperature difference we see between model runs with and without radiation (red line). This suggests again that radiation issues are not the main driver of this error.

In summary, we agree that uncertainties related to solar radiation should be more clearly mentioned and we will adapt the manuscript accordingly by summarizing the discussion above in the supplementary material and referring to it in Sections 3.2, 3.3 and 4.1.

[Figure]

Figure 7 – Lonza catchment (red) and river course (blue) with DEM added on top. Source: Swisstopo

[Figure]

Figure 8 – Top: Measured (black) and simulated water temperature (red with shortwave radiation, green without shortwave radiation) for the Alpine Lonza catchment. Bottom: Difference between measured and modeled (with shortwave radiation) water temperature (black), and difference between water temperature modeled with and without shortwave radiation (red).

Finally, the hypothesis suggested by the reviewer of the HSPF approach being the cause of the summer overestimation of the water temperature in Alpine catchments remains. Expressed in other words, this is already what we state in the paper. Indeed, when considering that the main explanation of the problem is the missing cold-water advection, and knowing that this is not considered in the HSPF approach, we realize that this could be an explanation for (a part of) the problem. To show this, the HSPF solver has been rewritten in R outside of the model and run over the Landwasser catchment using the mean air temperature over the catchment (computed from the grid output of Alpine3D). The results are shown in Figure 9. We clearly see that the peaks of high-water temperature in the river in summer correspond to peaks in the temperature of the HSPF outflow. This figure will be added in the supplementary material in Section S7 where we present the energy fluxes.

The issue of high-water temperature is extensively discussed in this review response. We thank the reviewer for the various hypotheses suggested. We performed an additional analysis for better identifying this issue for the purpose of this reply, completing the already extensive description in the paper. The new hypothesis tested here brings us however to the same conclusion as in the paper: the issue arises from not accounting for some cooling processes in the headwater regions for the water in the soil reservoirs. Snow and glacier melt (likely also melting permafrost) are the main candidates after having excluded many other hypotheses. This problem is well recognized in the paper and discussed in a transparent manner. We also provide some evidence of the robustness of the climate change simulations performed despite this error. It is shown in our work that the development of a new, more accurate scheme for dealing with water temperature in the soil reservoirs is important for Alpine regions, but this is beyond the scope of this paper.

We agree that several clarifications are needed in the manuscript; most of them have been suggested above and will be implemented in the manuscript as described.

The content of this answer has been added to Section S9 and its essence is incorporated in Section 4.2.2 (p.16 l.27-31 and p.17 l.3-13).

*Table 1: I recommend including some metrics of dominant land cover in this table (e.g., %forest, %agriculture, %urban, %lake, %rock/meadow).*

Thank you for the suggestion, we'll add it.

Table 1 has been updated as suggested. The text of Sections 2.1 and S3 along with Table S3 have been updated accordingly.

*Figure 1: Perhaps distinguish between 'lowland' and 'alpine' catchments using colour?*

Thanks for the suggestion, we'll use two different colors and re-arrange the numbering to be able to split the catchment list on the right between lowland and Alpine catchments.

Figure 1 has been updated following the reviewer's proposition.

[Figure]

Figure 9 – Top: mean air temperature over the Landwasser catchment and outflow water temperature from the HSPF scheme when forced with the mean air temperature (and the parameters calibrated over the Lanswasser catchment presented in Table S8 of the supplementary material). Bottom: difference between simulated and measured water temperature for the Landwasser catchment.

**REPLY TO REVIEWER 2**

Dear Reviewer,

First of all, we would like to thank you for this clear and helpful review. We are very well aware that this review has been a significant time investment and therefore especially appreciate the reviewer's feedback and commitment. We provide our detailed answers and explanations below and hope that these address and clarify the reviewer's comments and questions.

Reviewer comments are repeated in *italic*, author replies in regular font.

Best regards,

Adrien Michel, on behalf of the author team.

*The manuscript provides a thorough investigation of modeled future temperatures in Swiss streams. Methods are well detailed and simulated streamflow for historic and future conditions are exhaustively detailed.*

*My main comments involve the length of the manuscript and the primary messages. The manuscript is almost too long, with certain side analyses partially detracting from more central messages of the manuscript.*

*On a related note, the abstract itself predominantly focuses on the future simulations (which – as admitted in the manuscript – have some potential limitations) while neglecting what I see as the more fundamental insights into hydrological process and the sufficiency of model structure. The manuscript had more nuance and deeper investigations into process than I was led to believe by initially reading the abstract.*

*Thus, I have two specific thoughts:*

1. *Possibly move Figure 7, 8, and 9 and some accompanying text to Supplemental section*

We agree that the paper is rather long as it accommodates a substantial number of results. We will do our best during the revision process of the manuscript to shorten it. Figures 7, 8, and 9 could indeed be moved to the supplementary material along with part of the discussion only keeping the main message in the text. Reviewer 1 also suggested moving Section 5.3 to the supplementary material. We will consider these suggestions during the revision with the objective to shorten the paper and give the main messages better visibility.

[All numbering in this answer refers to the numbering in the original manuscript]

We made significant efforts to shorten the manuscript. Despite many additions and clarifications included following suggestions of the two reviewers, the text has been shortened by 10%. We also reduced the number of Figures from 12 to 8 (despite adding a completely new figure) and removed one table. We followed the suggestion of removing Figure 7, while combining Figures 8 and 9 in one new figure. We estimate the latter very relevant since they illustrate the concrete impacts of water temperature

increase. Figures 3, 4 and 6, along with Sections 4.2 and 4.3 (old numbering) have been moved to the supplementary material, and Section 3.4 has been removed.

Regarding Section 5.3, we prefer to keep it since it is a relevant piece of information to explain the difference between the temperature rise expected on the Swiss Plateau and in the Alpine catchments. However, we removed the correlation analysis spread between Section 5.1 and 5.2 and Table 6 since it was partly redundant with Section 5.3. Section 5.3 has been revised significantly to better integrate in the paper and show it's added value.

*Rewrite abstract to better emphasize insights into appropriateness of model structure and reduce emphasis on summary of future simulations. Similarly, the conclusions section could also benefit from some shifting of prioritization of messages. In particular, there should be specific mention of that model does not allow for direct input of melt water into streams and that this led to overestimate of warming under historical conditions (but is believed to be less of an issue in the future as snow diminishes).*

Thank you for this suggestion. Indeed, the paper is discussing quite extensively the models and their limitations. While the title gives some indication, this part is absent from the abstract as well as the conclusions. We definitely agree that these aspects should be included. Here is how we intend to adapt the abstract, showing additions (in green) and deletions (in red):

[revised manuscript text omitted]

Based on our modelling results, expected CC impacts on river water temperature in Switzerland can be summarized as follows: […]"

The abstract has been replaced by the one suggested above. The conclusion has been significantly edited and shortened.

---

## Author Response (AR2)

Dear editor, Dear reviewer,

We thank you very much for your constructive and instructive comments. We really appreciate the time taken by the reviewer to provide comments and suggestions on how to improve the presentation of our results. The deeper scientific discussion on the summer water temperature overestimation issue also helped us to reconsider the interpretation of the results and to partly reformulate the confidence related to some of our results.

Please find below a detailed answer to all the points raised in the review. We hope that these answers, together with the corresponding modifications of the manuscript, will clarify and resolve the remaining issues identified by the reviewer.

In the following, reviewer comments are repeated in *blue italic* and, author replies in regular font. Section and figure numbers refer to the revised version of the manuscript if not stated otherwise.

Best regards,

Adrien Michel, on behalf of the authoring team.

**General note**

As mentioned in a separate email to the editor, we found a minor error in the computation of the seasonal means in the previous version of the paper. The error was that first, monthly averages were computed, and then, seasonal averages were computed from the monthly averages. This has been corrected now by computing seasonal averages from all the days constituting the seasons. Changes in the results are marginal and do not affect any of the paper's outcome and analysis, but explain why some plots and tables in this new version have been updated and slightly differ from the previous version of the manuscript.

**Editor comment**

*Dear Authors,*

*Thank you for the resubmission of your manuscript. The manuscript has received one re-review, and the reviewer has raised several points they hope you can address through an additional revision. In particular, they recommend rewriting and reorganizing portions of the results and discussion, to ensure limited repetition and improve overall flow of the manuscript. They also raise an important point: the need for a clear statement of objectives in the introduction.*

Following the reviewer's suggestion, the manuscript has been substantially reorganized around two clear objectives which are now more prominently stated in the introduction. This resulted in a shorter (word count educed by 18%) and better organized manuscript (the results and discussion have been completely reorganized) that, at the same time, provides more pertinent information than our first version. Again, we thank the reviewer for their insights and useful suggestions which helped to improve the manuscript.

*They also raise some strong points regarding future simulation with model results that over-predict during summer periods. I encourage you to be forthright with these limitations in your revised manuscript, and to address some of the questions from the reviewer below regarding this overprediction (recognizing as well that you have made a major effort to reduce the length of this manuscript in your previous revision).*

The question of water temperature overestimation in Alpine catchments is discussed below. We provide evidence supporting our hypothesis of a missing cooling mechanism in the paper and in our answers. The presence of the observed issues is now also well discussed as one of the limitations of this work.

Following the concerns of the reviewer, and re-evaluating our results, we lowered the confidence in the results for the Alpine catchments in summer and removed some related part of the analysis. All other results (lower elevation catchments, the metrics used there to assess the impact of CC, and the general pattern of warming in the Alpine regions) are shown to be robust and bring, in our opinion, relevant new insights.

*I look forward to seeing an updated version of your manuscript!*

Please find the new substantially revised version in the attachment.

**Reviewer comment**

*I reviewed the initial submission of this manuscript. I appreciate the detailed responses to the reviewer comments and actually found many of the responses around model decisions and approaches engaging and insightful. I read the responses prior to reviewing the new version and I was excited to read how the authors had incorporated this material into the revision.*

Thank you for acknowledging the work carried out in the previous revision.

*Personally I find the model evaluation steps and decisions much more interesting than the climate change simulations. I think we learn more about our systems through testing models and understanding why and where they succeed and fail. In contrast, climate change simulations really amount to a glorified sensitivity analyses that often doesn't provide many new insights into system behaviour or advancing our scientific knowledge. Based on the reply to comments, I was expecting to see more of the 'learning from models' piece in the manuscript. I think some of that discussion is there, but unfortunately, I continue to find the new version of the manuscript difficult to follow and overwhelming, which ultimately reduces the impact of this study. I provide some suggestions on how the presentation could be improved.*

We realized and agreed that the paper still had potential for being streamlined, shortened, and sharpened. The "impact studies" versus "model mechanisms" question is discussed below.

*I also have a remaining concern about some of the arguments around the model over-estimation for summer temperatures for the Alpine catchments. I detail these comments below, followed by some more specific comments.*

*1. Organization and focus of the paper*

*I continue to struggle getting through this manuscript and I've spent a bit of time reflecting on why that is. There is an immense amount of material presented which partly contributes to the confusion, but in thinking about this more, I suspect some serious efforts in restructuring and adding focus to the manuscript will ultimately make it easier to read and communicate the key points of this study. Here are a few suggestions:*

*- Some of the result sections contain methodological details, as well as discussion points. This makes it difficult to follow some of the lines of reasoning, as well as getting the key points from the study. In addition, the reader is constantly referred to different sections ('see Section X', 'presented below', 'details below', etc.), as well as being referred to the (extensive) supplemental material and details in other papers – which makes it all challenging to follow, even on the third read through, let alone the first. I think by keeping introduction, methodology, results and discussion material in their own sections would improve the logical flow of the paper. I*

*also think the authors should spend some time creating a logical narrative throughout the manuscript that is sufficiently supported with standalone evidence and explanations without having the reader search all over for information. However, you don't want to include so much detail that it results in an overly long paper. I recognize this is a difficult balance to strike with the amount of modelling work done here, but the current version remains difficult to follow and digest.*

We agree with the reviewer's points, which are well justified. Following these suggestions, the paper has been substantially reorganized:

- In the introduction, we reformulated the objectives of the study (Page 3 Lines 3-9): *"The present study has two main objectives: i) Assess the ability of a physics-based model chain to simulate discharge and water temperature. This is achieved by using performance metrics over calibration and validation periods, and by assessing in how far the models are able to reproduce currently observed trends. ii) Investigate the impact of CC on river temperature. Despite the existence of extensive recent studies on discharge evolution under CC in Switzerland over a larger set of catchments (Brunner et al., 2019a, b; Muelchi et al., 2020, 2021), discharge is included in our analysis given the coupling of water temperature and discharge. For both objectives, the comparison of lowland versus Alpine catchments is one of the focal points of this research."*

- The modified Results Section presents the main findings of this study. First, we present the calibration results and the model performance (this is now a clear result according to objective 1), followed by the presentation of the impact study results.

- A newly added Discussion Section first concentrates on the discussion of the models' performance, in particular the problem in Alpine catchments, and then on the discussion of the impact of climate change.

- We completely reframed and shortened the conclusions to briefly summarize our main findings in view of the two principal objectives of the study.

We provide the manuscript with tracked changes, however, due to the major reorganization it would be endless to mention here all changes appearing in the track-change version. Note that the track-change fails sometimes to attribute deletions to the correct section. We therefore mention in Table 1 below the main modifications in the manuscript.

Overall, the revised version of the manuscript is 18 % shorter than the previous version (and 24% shorter than the first version). It is still a long paper, but this is necessary for reporting and discussing all relevant information. We think that the new organization improves the readability of this paper. By shortening and reorganizing the paper, many cross-references, which were another concern of the reviewer, have been removed.

Table 1 – Summary of the main reorganization and modifications of the manuscript.

| Section (previous version) | Section (new version) | Changes |
|---|---|---|
| Abstract | Abstract | Some details have been removed, others have been added, e.g. regarding the limitations. |
| 1. Introduction | 1. Introduction | Reorganized. Some details have been removed. The part about statistical vs. physics-based model has been shortened substantially. |
| 2. Data | 2. Data | Removal of some repetitions and cross-references. |
| 3. Models | 3. Models | Removal of some repetitions and cross-references, removal of some minor details. |
| 4. Models calibration and validation | Deleted | Section 4 became the "Results" Section. |
| 5. Climate change simulations – Results and discussion | 4. Results 5. Discussion | Section 5 was split in two separate sections (Results and Discussion). |
| 4.1 Calibration and validation setup | 3.4 Models calibration and validation | The technical procedure of the calibration is now described in Section 3.4, as part of the main "Model" Section. |
| 4.2 StreamFlow calibration and validation results | 4.1 StreamFlow calibration and validation results 5.1 Model chain performance | Former Section 4.2 is now part of the Results Section (4.1). The second subsection, about Alpine catchments, has been shortened to only presenting the results (including the summer problem). All discussion of the performances of the model and the issues has been moved to Section 5.1. The content has been edited to reflect the discussion below on the HSPF scheme and on the uncertainty. |
| 5.1 Swiss Plateau Catchments | 5.2 Climate change impact | The discussion of the CC impact, with the two indicators, has been moved to the Discussion Section. |
| 5.2 Alpine catchments | 5.2 Climate change impact | The discussion of soil temperature has been moved to the Discussion Section. The discussion on summer warming impacts has been shortened due to related uncertainty. |
| 5.3 Role of discharge variations for summer water temperature | 5.2 Climate change impact | The discussion of sensitivity of water temperature to discharge has been moved to the Discussion. Section. Only Plateau catchments are now discussed and, in less |

| | | detail, since this is not directly related to one of the two main objectives. |
|---|---|---|
| 5.3 Robustness, limitations and open questions. | 5.3 Limitations and open questions | Robustness is now discussed in Section 5.1 and Section 5.3 now focuses only on the other aspects. |
| Conclusion | Conclusion | Part about CC impact discussion has been removed, while part of model mechanism analysis has been enhanced, to better report on the study's two main objectives. |

*- The introduction could be substantially streamlined. It jumps around and lacks a logical flow. In addition, the study would be better contextualized with clear statements of the study objectives. Currently, the closest I could find to stated research objectives in the last paragraph of the introduction amount to (a) a statement about which models were employed in this study and (b) that these models were used to investigate the impact of climate change. More specific and testable objectives (or even hypotheses) would give the study greater focus and impact.*

The introduction has been reorganized profoundly to achieve a better logical flow. Details of lower importance have been removed. The research objectives are clearly stated upfront. We thank the reviewer for this comment which largely helped for the subsequent paper reorganization.

*- Some detailed editorial work by the authors is needed. The text throughout can be streamlined - such as removing sentences like Page 1 Line 3-4 and Page 1 Lines 8-9 ('This represents...'). Another example is that the paragraph on page 2 lines 16-21 could be removed or incorporated more succinctly into the opening paragraph. These are just examples and I encourage the authors to critically review the writing throughout to cut unnecessary words and sentences. In addition, the grammar could also be improved throughout.*

We removed many clauses or extra-verbosity in the manuscript (see track-changed version). We also checked the grammar of the document which, combined with the excellent language editing service freely available for Copernicus publications, should result in a good general grammar level for a scientific publication.

*A simple suggestion, but perhaps add a prefix to the catchment names to designate whether they are plateau or alpine sites (e.g., P-Suze, A-Lonza or something similar). Keeping track of these throughout the manuscript is difficult, particularly since they are not always grouped together in a logical way (especially in the supplemental document).*

We tried to follow the suggestion of adding prefixes, but we felt it did not lead to the desired clarity. We now reorganized the plots in Supplementary Material to always distinguish between Plateau and Alpine catchments, in an attempt to facilitate the reading. Also, the catchment type is now always mentioned in the Figure captions. In the main text, we now have distinct sections or paragraphs in the Result and Discussion Sections referring exclusively to one type of catchment.

*I struggle with the logic around continuing with the climate change simulations despite the over-prediction of summer stream temperature at the Alpine catchments. It seems that the authors are arguing that the missing cooling mechanism is conclusively known (cold water advection not captured by the HSPF scheme), hence why the model over-predicts - but that's okay, since that won't be an important process under future climate conditions. This seems like two big assumptions (the cold water advection term is the missing component and that it won't be important in the future)! Despite the multitude of graphs, I don't think there is one that shows future absolute SWE simulations (although there are percent change figures). Is some SWE simulated for future scenarios? If so, I would argue that even if the cold water advection is the missing term, it will remain a key missing process in these models that is potentially important for correctly simulating river temperatures in these systems; therefore, how much can we believe these predictions?*

Yes, absolute SWE is modelled in Alpine3D, see Figure 1 below (similar figures for all catchment and for the months of December, January and May have been added to the supplementary, Figures S81 to S86). The computation of change in SWE percent is based on these data. In the manuscript, it is now explicitly stated that this information is simulated in Alpine3D (Page 18 Lines 6-11). The other points raised are addressed below.

*I appreciate the systematic evaluation of the potential mechanisms that could be accounting for the over-prediction; however, without a way to check some of the internal processes (e.g., evaluating the estimates of incoming solar radiation reaching the stream surface, above-stream wind speeds and humidity, lateral inflow temperatures and magnitudes, potential influences of groundwater/hyporheic exchange), it seems very difficult to conclusively say that the limitation in the HSPF scheme is the single cause driving the model error. What about evaporative cooling - is the latent heat flux underestimated in summer (I believe the alpine rivers are more open and perhaps characterized by greater ventilation - perhaps this is not properly captured in the model)?*

From our detailed analysis (Section 5.1 and S9), we can exclude many potential causes. The reviewer suggests additional reasons. The fact that the model behaves correctly in the lowland Plateau catchments and in Alpine catchments except for some sudden warming in summer suggests that the physical processes shared between both groups of catchments are correctly represented. Errors related to in stream processes or due to input data error or scarcity are not expected to cause such errors, which occur during a (short) well-defined time. The sudden warming peaks erroneously simulated do not always occur at the same period of the year and do not depend on discharge (note that some of the variables mentioned by the reviewer, such as radiation, latent heat fluxes, or lateral energy fluxes are shown in Figures S52 to S58 and discussed in Section S9). Conversely, the warming peaks are strongly correlated to peaks produced by the HSPF parametrization of input water temperature (see Figure S59).

This correlation indicates that it is very likely that the error stems from an overestimated energy input from the HSPF scheme. Considering these sudden energy inputs as correct, while then missing some "in-stream" mechanisms that would dissipate the extra energy only during these short-lived events, seems quite unlikely.

The HSPF approach, while being probably the best simple approach available (see our comparison with other approaches and the study of Leach et al., 2015), still remains relatively basic given its sole dependence on air temperature. In the Plateau catchments, air temperature is a good proxy for soil temperature and thus most likely also for subsurface flow water temperature (very few observations available), and leads to very good results. In the Alpine catchments, as we state on Page 22 Line 32-Page 23 Line2, interaction with a complex water table topology and snow/ice melt-induced effects are not considered by the model.

[Figure]

Figure 1 – Average snow water equivalent (SWE) and ice water equivalent (IWE) over the Inn catchment for the month of May. Maps show the average between the 8 model chains used (see Table 2 in the paper). For glaciated pixels, SWE and IWE are summed. Top: RCP2.6. Bottom: RCP8.5. Left: Reference period (1990-2000). Right: End of the century period (2080-2090).

During the early melt season, the air temperature is low (compared to summer) and close to or below subsurface soil temperature; the temperature of water feeding the stream will thus be close to air temperature independent of the actual flow paths (surface, subsurface) and independent of its recent origin from snowmelt or from rain; the HSPF approach is thus leading to correct results despite missing the actual surface/subsurface water exchange mechanisms and in particular by missing fast cold melt water advection at the surface.

The HSPF approach only fails when the air temperature is high, i.e. during periods when the HSPF approach produces large heat input whereas fast surface flow processes could lead to cold water advection from melt. This is another argument to identify the missing subsurface flow paths of meltwater as a potential culprit for the water temperature overestimation during summer. New evidence of the impact of cold advection by ice melt on water temperature has been recently published by Du et la. (2021). This reference has been added in the Discussion Section.

We do not expect the proposed cooling mechanisms (snow and ice melt or local groundwater interaction), if included in the model on top of the HSPF formulation, to directly suppress the large warming peak. We argue here that a completely different approach including these mechanisms should be tested to assess the ability of the model to correctly simulate the water temperature in Alpine catchments (Page 23 Lines 10-11 and Lines 10-20, see also the answer to the next reviewer point for additional elements). However, we also clearly acknowledge that these proposed mechanisms are uncertain and would need further investigation (Page 23 Lines 21-23).

This discussion clearly falls under the first objective of the paper and is extensively treated in Section 5.1 which integrates new elements from the discussion above. We clearly point out the limitation of the HSPF approach and suggest mechanisms that should be accounted for to

improve hydrological models. Given the length of the paper, this discussion is not exhaustive and does not cover every aspect in all detail. We consider a systematic solution of the problem of overestimated summer water temperature in Alpine catchments to be beyond the scope of this study.

In the initial version of the manuscript, we stated that snowmelt was the major (and probably only) problem and, as it will be of lesser importance in the future, that it would be a minor issue for the impact assessment. We reconsidered this statement in view of the identification of the mechanisms discussed above, leading to some implications on the presentation and interpretation of the results (see below). However, the reduction of snow and glacier cover, now also shown in the new maps in Figures S81 to S86, and its potential impact on water temperature are still discussed in the revised manuscript (Page 23, Lines 24-32).

*What about errors in input data? For example, Figures S36-S46 highlight some interesting differences between the measured and CC scenario precipitation estimates. It seems for many of the streams that exhibit over-predictions in summer stream temperature are also associated with under-estimates of summer precipitation (and discharge), such as Lutschine (Figure S44). Is precipitation under-estimated in these alpine catchments (which is often an issue in high elevation catchments) driving the over-prediction of river temperature?*

When forced with CC scenarios during historical periods, discharge simulations show indeed lower performance in Alpine catchments, which is mentioned in the text (Page 15 Lines 10-15). However, the error in temperature does not correlate with the period of maximum discharge underestimation (see e.g. Figures S42 and S45) and the "incorrect" summer warming also appears in catchments where summer discharge is correctly simulated or even overestimated during the summer season when forced with CC scenarios (see e.g. Figures S43 and S44). In addition, the overestimation is also observed when forced with measurements and not only when forced with CC scenarios (Page 22 Lines 15-19):

*"These overestimations do not appear during all summers and there is no temporal coincidence between the instances of temperature overestimation and low discharge conditions. Furthermore, only two of the rivers concerned with this overestimation problem (the Inn and the Landwasser) show a correlation between river temperature and discharge errors (Figures S21, S23, S25, S27, and S29), suggesting that the underestimation of summer discharge cannot explain the overestimation of river temperature."*

The same conclusion arises from a comparison of the transient discharge and water temperature measurement compared to simulation, e.g., in Figures S20 and S26. Finally, a general bias in discharge prediction in summer would lead to a general bias in water temperature throughout the summer season, but not to sudden warming as simulated here.

*Overall, this seems to be a major challenge and undermines the confidence in the climate change scenarios. Similar to my point in the above, I think focusing on this model failure is far more interesting than whether the model simulates X degrees celsius of warming in 50 years, so I don't see this model limitation as barrier to publication. Instead, a repositioning of the manuscript to focus on these model failures and expanding the possible causes of these over-predictions (beyond what is already done), could highlight some important areas of future study to improve our simulations of stream temperature in these types of environments. I realize these types of climate change impact simulations are published all the time, so I'll understand if the authors would like to stick with that focus of the manuscript, but personally I think focusing more on exploring the model failure could make for a more satisfying study.*

We understand the opinion of the reviewer about the added values of the "model mechanism" studies with respect to climate impact studies. However, impact studies are of great value as a basis for decisions or further studies (for instance on the impact on the aquatic biota). At the same time, model mechanisms have also been a focal point of this study, cause for the length

and earlier structural issues of this paper. These issues have now hopefully been eliminated thanks to the reviewer's suggestions.

We think that simultaneously addressing the model mechanism question and the climate change impact is beneficial. In many recent hydrological impact studies, the model validation part is discussed only very briefly. If we had decided to not go into details, and only show the RMSE results for the Alpine catchments, which are similar to the ones in the Plateau catchments, the summer overestimation would have been unnoticed. This is in no way an argument to downplay the observed issue in this paper, but rather a point to justify a detailed model discussion in impact studies, even if this comes at the price of having a rather lengthy paper.

We think that the revised paper now has a much better balance of the two main objectives and the results about the model mechanism (in our case, the overall good performance, except the issues in Alpine catchments during summer) receive the necessary attention. In particular, the model performance is now discussed as part of the Results Section and the discussion of the model problem is placed in the Discussion Section.

We now address the reviewer's remark concerning the impact study, i.e., the question of treating the summer overestimation and confidence in the values obtained. The main argument given in the initial version was that the identified mechanism will decline and disappear in the future. We acknowledge that our two points are assumptions: (1) correct identification of the mechanisms, and (2) that their impact will actually decrease in the future.

As discussed above, we have strong arguments identifying the HSPF scheme as the culprit but cannot say whether the lacking mechanism in the HSPF scheme is snow/ice melt related (and thus expected to decrease in the future), or some other mechanism such as groundwater interaction (for which it is unclear how it will evolve in the future). This is now stated on Page 23 Lines 16-20, and related statements concerning the (un)certainty we put on these results (Alpine catchments during summer) are adjusted accordingly. Specifically, the manuscript has been modified as follows:

- We shortened the description of the analysis of the summer results in Alpine catchments (Page 19) and clearly mention the related uncertainty and refer to the Discussion Section (Page 20 Lines 1-2),

- When discussing the summer overestimation, we clearly state the related uncertainty and say that our main result regarding the summer temperature in Alpine catchment is rather the identification of the issue and the proposed mechanism to explain it (Page 23, Line 21-23),

- In the Discussion Section, we stress that the actual numbers predicted come with a given uncertainty, and emphasize more on the elements that will impact water temperature in the future in Alpine regions (Page 24 Lines 21-31),

- In the discussion of the sensitivity of water temperature to discharge, we state that the difference observed between Swiss Plateau and Alpine catchments would be worth investigating in more detail, and that for now the uncertainty of summer water temperature in Alpine catchments prevents us from drawing strong conclusions (Page 26 Line 10-15).

Despite this reduced confidence in some specific results presented in the updated study (one season in one of the two types of catchments studied), we have strong confidence in all other results presented. We think that the majority of these results certainly will be of interest to our peers, e.g., for the study of their impact on ecosystems in rivers or for the interaction with lakes downstream of rivers.

*Some specific comments:*

*Page 1 Line 8: This sentence could be removed ('This represents...').*

Done.

*Page 2 Line 2: I don't think this sentence is needed here.*

Indeed, especially in the revised version this is even less relevant. We removed it.

*Page 2 Line 9: Do you mean 'along with' instead of 'along to'?*

Yes, this has been corrected.

*Page 2 Line 22-27: The logical development of this paragraph is confusing. It starts with a statement about river temperatures impacting groundwater temperatures, but then jumps into how precipitation patterns will impact groundwater thermal regimes and then concludes with this being a rationale for studying future changes in stream temperature. Some of these points are important, but how they are developed and linked could improved.*

The whole introduction has been reworked to obtain a better logical flow.

*Page 3 Lines 1-2: Could you briefly expand on what differences were found in that study instead of just stating there were differences?*

This has been added, see Page 2 Lines 23-24: "So far, the warming rate of rivers in Swiss Plateau catchments has been almost twice that of Alpine catchments."

*Page 3 Lines 7-8: This doesn't seem like the most appropriate reference here as the Horton et al preprint hardly talks about river temperature models.*

Indeed, the reference has been removed.

*Page 4 Line 6: Could you explain to non-Swiss readers why having both Swiss Plateau and Swiss Alps catchments is important?*

This has been briefly explained, see Page 3 Lines 10-12: "The focus is on Switzerland, a country presenting a wide topographic heterogeneity leading to different discharge and thermal regimes between the lowland Swiss Plateau regions, where the hydrological cycle is mainly precipitation driven, and the high altitude Alpine regions, where snow and glacier melt play an important role."

*Page 5 Lines 3-4: Do you mean the results of Epting et al 2021 were used in the current study or that the current study results were used in Epting et al 2021?*

The results of the current study have been used in the study of Epting et al., this has been clarified (see Page 3 Lines 28-30).

*Page 12 Line 15: 'first' instead of 'fist'*

Corrected.

*Page 13 Figure 3: Spelling mistake in the legend.*

Corrected, thank you for catching it.

*Page 15 Lines 3-4: Are these sorts of influences (e.g., cement factory effluent) common near the observation points in these catchments? I thought these catchments were selected because they had minimal anthropogenic influences?*

This is the only catchment with this kind of disturbances in our selection. Indeed, we tried to select catchments with minimal disturbance. But we also selected catchments suitable for the

groundwater study of Epting et al. (2021). The Suze is interesting in terms of groundwater, and has been retained despite some perturbations. It also serves as a good test case for a model setup in a very complex environment, i.e., a karstic region.

*Page 27 Lines 13-14: The Du et al study was conducted in Canada, not the USA.*

Thank you, corrected.

*Page 28 Lines 16-17: But on Page 4, Lines 8-11 it was stated that these study catchments were selected based on criteria (e.g., no large lakes, no anthropogenic disturbances, no dams) that other Swiss catchments didn't meet; therefore, are they actually representative, since it's well established that lakes, dams and anthropogenic disturbance can have profound influence on river thermal regimes?*

We changed the sentence to: "The studied catchments can be assumed to be representative of **undisturbed** Swiss catchments in general"; this clarification is indeed needed here.

*Figures S52-S58: Are these daily or hourly values presented in the graphs? If hourly, the upper range in incoming solar radiation seems low around the summer solstice.*

All values shown in these figures are daily values, this is now clarified in the captions.

**References**

Brunner, M. I., Björnsen Gurung, A., Zappa, M., Zekollari, H., Farinotti, D., and Stähli, M. (2019a). Present and future water scarcity in Switzerland: Potential for alleviation through reservoirs and lakes, Science of The Total Environment, 666, 1033–1047, DOI: 10.1016/j.scitotenv.2019.02.169.

Brunner, M. I., Farinotti, D., Zekollari, H., Huss, M., and Zappa, M. (2019b) Future shifts in extreme flow regimes in Alpine regions, Hydrology and Earth System Sciences, 23, 4471–4489, DOI: 10.5194/hess-23-4471-2019

Du, X., Silwal, G., Faramarzi, M. (2021) Investigating the impacts of glacier melt on stream temperature in a cold-region watershed: coupling a glacier melt model with a hydrological model, Journal of Hydrology, 127303, DOI: 10.1016/j.jhydrol.2021.127303.

Leach, J. A., & Moore, R. D. (2015). Observations and modeling of hillslope throughflow temperatures in a coastal forested catchment. Water Resources Research, 51(5), 3770-3795, DOI: 10.1002/2014WR016763, 2015.

Muelchi, R., Rössler, O., Schwanbeck, J., Weingartner, R., and Martius, O. (2020): Future runoff regime changes and their time of emergence for 93 catchments in Switzerland, Hydrology and Earth System Sciences Discussions, 2020, 1–25, DOI: 10.5194/hess-2020-516

Muelchi, R., Rössler, O., Schwanbeck, J., Weingartner, R., and Martius, O. (2021): Moderate runoff extremes in Swiss rivers and their seasonal occurrence in a changing climate, Hydrology and Earth System Sciences Discussions, 2021, 1–28, DOI: 10.5194/hess-2020-667